# Mitigating Goal Misgeneralization via Minimax Regret

## Abstract

Robustness research in reinforcement learning often focuses on ensuring that the policy consistently exhibits capable, goal-driven behavior. However, not every capable behavior is the intended behavior. *Goal misgeneralization* can occur when the policy generalizes capably with respect to a 'proxy goal' whose optimal behavior correlates with the intended goal on the training distribution, but not out of distribution. Though the intended goal would be ambiguous if they were perfectly correlated in training, we show progress can be made if the goals are only *nearly ambiguous*, with the training distribution containing a small proportion of *disambiguating levels*. We observe that the training signal from disambiguating levels could be amplified by regret-based prioritization. We formally show that approximately optimal policies on maximal-regret levels avoid the harmful effects of goal misgeneralization, which may exist without this prioritization. Empirically, we find that current regret-based Unsupervised Environment Design (UED) methods can mitigate the effects of goal misgeneralization, though do not always entirely eliminate it. Our theoretical and empirical results show that as UED methods improve they could further mitigate goal misgeneralization in practice.

## 1 Introduction

As reinforcement learning (RL) is increasingly applied in complex, open-ended, real-world environments, it becomes infeasible for training to comprehensively cover all situations that an agent will face in deployment. A particular challenge arises when insufficiently diverse training creates a 'proxy goal' that, compared to the true goal, induces similar behavior during training but radically different behavior out of distribution. In the face of proxy goals, standard RL methods sometimes find a policy that internalizes the wrong goal, completely ignoring the true goal in favor of capably pursuing the proxy goal out of distribution (Langosco et al., 2022; Shah et al., 2022). This phenomenon, known as goal misgeneralization, is a pressing problem in assuring the safety of RL agents (Ngo et al., 2023).

If the true goal and the proxy goal are perfectly correlated in the training distribution, then goal generalization comes down to the algorithm's inductive biases. However, we find that we can make progress on goal misgeneralization regardless of the algorithm's biases by working in a relaxed setting where the training distribution is only *nearly ambiguous*, providing a weak training signal in favor of optimizing the true goal. In particular, we model a complex environment as comprising a series of *levels* (Cobbe et al., 2020; Kirk et al., 2023) which can be designed during training (Dennis et al., 2020). We assume that while most training levels leave the true goal ambiguous, a small proportion of levels are available to disambiguate the goals by incentivizing different behavior. In this setting, we show that the standard RL training method of domain randomization (DR; Tobin et al., 2017), which optimizes for expected return on the training level distribution, may still induce goal misgeneralization if the proportion of disambiguating levels in training is sufficiently small.

However, we observe that even when following the proxy goal is approximately optimal in terms of maximum expected return on the training distribution, it is clearly sub-optimal in terms of minimax expected regret (MMER; Savage, 1951), since it leaves the true goal unfulfilled in disambiguating levels. We hypothesize that MMER-based training methods such as unsupervised environment design (UED; Dennis et al., 2020) should naturally mitigate this kind of goal misgeneralization, anticipating a possible distribution shift and amplifying the weak training signal from disambiguating levels. In this paper, we contribute the following theoretical and empirical results in support of this hypothesis:

- In Section 3, we formalize our setting involving a nearly-ambiguous training distribution and a distribution shift to a test distribution involving mostly disambiguating levels.
- In Section 4.1, we prove our main theoretical result, that in the face of such a distribution shift, (A) approximate DR is susceptible to goal misgeneralization, but (B) approximately optimizing MMER is robust to goal misgeneralization.
- In Section 4.2, we introduce an abstract model of the consequences of goal misgeneralization in RL based on the existence of a finite resource that can be allocated to either the true goal or the proxy goal. This provides a broad sufficient condition for our main result to hold.
- In Section 5, we complement our theoretical findings with experiments in custom JAX-based grid-world environments with procedurally-generated levels exhibiting proxy goals. We show that two UED methods—namely (PLR$^{\perp}$; Jiang et al., 2022) and (ACCEL; Parker-Holder et al., 2023)—are significantly more robust to goal misgeneralization than DR.

These results support regret-based UED methods as a promising approach for safe goal generalization. We note that this approach requires the ability to design training levels, but this requirement is often satisfied in practice, having a simulator in sim-to-real transfer (Tobin et al., 2017; Peng et al., 2018; Kumar et al., 2021; Makoviychuk et al., 2021; Muratore et al., 2022; Ma et al., 2024), having a generative environment model (Bruce et al., 2024), or having a world model (Ha & Schmidhuber, 2018; Hafner et al., 2019; Schrittwieser et al., 2020; Hafner et al., 2023; Valevski et al., 2024).

## 2 PRELIMINARIES

### 2.1 UNDERSPECIFIED MARKOV DECISION PROCESSES

A reward-free *underspecified Markov decision process* (UMPD) is a tuple $M = \langle A, \Theta, S, \mathcal{I}, \mathcal{T} \rangle$ where $A$ is the agent's action space, $\Theta$ is the space of the free parameters of the environment, $S$ is a state space, $\mathcal{I} : \Theta \to \Delta(S)$ is an initial state distribution, and $\mathcal{T} : \Theta \times S \times A \to \Delta(S)$ is a conditional transition distribution. An agent's behavior in an UMDP is represented by a *policy*, a conditional action distribution $\pi : \Theta \times S \to \Delta(A)$. We denote by $\Pi$ the set of all policies. Given a level $\theta \in \Theta$ we have a reward-free MDP $M_\theta = \langle A, S, \mathcal{I}(\theta), \mathcal{T}(\theta, -, -) \rangle$ and an agent's level-specific policy $\pi(\theta, -)$. We omit $\theta$ when not relevant or clear from the context. Together with the initial state distribution and transition distribution a policy $\pi$ induces a distribution over *trajectories* $\tau = (s_0, a_0, s_1, a_1, \ldots)$ with $s_0 \sim \mathcal{I}(\theta)$, $a_t \sim \pi(\theta, s_t)$, and $s_{t+1} \sim \mathcal{T}(\theta, s_t, a_t)$. Given a full trajectory $\tau = (s_0, a_0, \ldots)$ we denote by $\bar{\tau} = (s_0, s_1, \ldots)$ the corresponding *state trajectory*. A given policy similarly induces a distribution over state trajectories. We denote by $\mathrm{T}$ the set of state trajectories with positive probability under some policy.

Given a reward-free UMDP $M$ and a level $\theta \in \Theta$, let $R : S \times A \times S \to \mathbb{R}$ be a reward function, and let $\gamma \in [0, 1]$ be a discount factor. Together, $M$, $R$ and $\gamma$ make a regular UMDP, but we define the reward functions separately so that we can more easily talk about one environment given different reward functions. We define the *return (with respect to R)*, $U^R(\tau)$, as the discounted reward collected across trajectory $\tau$, $U^R(\tau) = \sum_{t=0}^{\infty} \gamma^t R(s_t, a_t, s_{t+1})$. We assume reward functions are normalized such that the returns lie in the range $[0, 1]$. We define the expected return of a policy $\pi$ under $R$ as $U^R(\pi; \theta) = \mathbb{E}_\tau \left[ U^R(\tau) \right]$ with the expectation taken over the distribution of trajectories induced by the policy $\pi$ and the level $\theta$. Given a distribution over levels $\Lambda \in \Delta(\Theta)$, we further define the expected return of a policy over a UMDP as $U^R(\pi; \Lambda) = \mathbb{E}_{\theta \sim \Lambda} \left[ U^R(\pi; \theta) \right]$.

Let $\varepsilon \geq 0$ be an approximation threshold. Given a reward function $R$ and a level $\theta$ we define the *approximately optimal policy set* as $\Pi_\varepsilon^\star(R, \theta) = \{ \pi \mid U^R(\pi; \theta) \geq \max_{\pi'} U^R(\pi'; \theta) - \varepsilon \}$. Analogously, given a distribution $\Lambda \in \Delta(\Theta)$, define $\Pi_\varepsilon^\star(R, \Lambda) = \{ \pi \mid U^R(\pi; \Lambda) \geq \max_{\pi'} U^R(\pi'; \Lambda) - \varepsilon \}$.

### 2.2 UNSUPERVISED ENVIRONMENT DESIGN

The standard method of training in an UMDP is to use *domain randomization (DR)*, training on levels sampled independently from a fixed level distribution. This leads to a policy that (approximately) maximizes the expected return given that distribution. Formally, given a fixed training distribution $\Lambda \in \Delta(\Theta)$ and reward function $R$, DR seeks to maximize the objective $\mathbb{E}_{\theta \sim \Lambda} \left[ U^R(\pi; \theta) \right]$. The set of approximately optimal DR policies, denoted $\Pi_\varepsilon^{\mathrm{DR}}(R, \Lambda)$, is simply $\Pi_\varepsilon^\star(R, \Lambda)$.

*Unsupervised environment design* (*UED;* Dennis et al., 2020) proposes training in UMDPs via a two-player game, where an agent is trained on levels selected by an adversary. In *regret-based UED,* the agent tries to minimize *expected regret* while the adversary tries to maximize it, where the *regret* on a level is the shortfall of return achieved by the policy compared to an optimal policy, $\mathcal{G}^R(\pi; \theta) = \max_{\pi'} U^R(\pi'; \theta) - U^R(\pi; \theta)$. This is a zero-sum game, and at the Nash equilibrium, the agent plays a minimax expected regret (MMER) policy, achieving $\min_{\pi \in \Pi} \max_{\Lambda \in \Delta(\Theta)} \mathbb{E}_{\theta \sim \Lambda} \big[ \mathcal{G}^R(\pi; \theta) \big]$. Note that MMER is distinct from maximizing minimum expected *return* (cf. Dennis et al., 2020).

Given a policy $\pi$, the adversary's best response set $\mathrm{BR}(\pi; R) \subseteq \Delta(\Theta)$ is defined as

$$\mathrm{BR}(\pi; R) = \arg\max_{\Lambda \in \Delta(\Theta)} \mathbb{E}_{\theta \sim \Lambda} \big[ \mathcal{G}^R(\pi; \theta) \big] \tag{1}$$

The set of approximately optimal policies under the MMER objective is therefore defined as

$$\Pi_\varepsilon^{\mathrm{MMER}}(R) = \left\{ \pi^{\mathrm{MMER}} \,\middle|\, \pi^{\mathrm{MMER}} \in \Pi_\varepsilon^\star(R, \Lambda^{\mathrm{MER}}) \text{ and } \Lambda^{\mathrm{MER}} \in \mathrm{BR}(\pi^{\mathrm{MMER}}; R) \right\} \tag{2}$$

Note that while the policy is approximately optimal, the adversary's response is optimal.

## 3 PROBLEM SETTING

When the true goal and the proxy goal perfectly correlate on the entire training distribution, it is impossible to distinguish between polices optimizing for either goal. Hence, we propose to study goal misgeneralization in a relaxed setting with a *nearly ambiguous* training distribution—we assume that a small subset of training levels provide a weak training signal that disambiguates the true goal from the proxy goal. This relaxation mirrors the assumptions made in previous work on spurious correlations in supervised learning (Liu et al., 2021; Zhang et al., 2022).

To formalize our setting, we next introduce and define the concepts of proxy goals, what it means for a level to be disambiguating or ambiguating, and of a $C$-distinguishing distribution shift.

**Definition 3.1** (Proxy goal). *Given a reward-free UMDP $M$, a level distribution $\Lambda \in \Delta(\Theta)$, and a (true) reward function $R$, we say a reward function $\tilde{R} : S \times A \times S \to \mathbb{R}$ is a* proxy goal *if there exists a* proxy policy *$\tilde{\pi}$ which is approximately optimal with respect to both the given goal and the proxy goal: $\exists \tilde{\pi} \in \Pi_\varepsilon^\star(R, \Lambda) \cap \Pi_\varepsilon^\star(\tilde{R}, \Lambda)$.*

**Definition 3.2** (Ambiguating level). *Given a reward-free UMDP and a pair of reward functions $R$ and $\tilde{R}$, a level $\theta \in \Theta$ is* (perfectly) ambiguating *if all optimal policies with respect to either reward are approximately optimal with respect to the other as well, that is,*

$$\Pi_0^\star(R, \theta) = \Pi_0^\star(\tilde{R}, \theta)$$

**Definition 3.3** ($C$-disambiguating level). *Given a reward-free UMDP, a pair of reward functions $R$ and $\tilde{R}$, and a constant $C > 0$, a level $\theta \in \Theta$ is $C$-disambiguating *if all policies that are optimal with respect to $\tilde{R}$ achieve $C$-sub-optimal return with respect to $R$, that is,*

$$\Pi_0^\star(\tilde{R}, \theta) \cap \Pi_C^\star(R, \theta) = \emptyset$$

We note that some levels may be neither perfectly ambiguating nor $C$-disambiguating for any $C > 0$.

**Definition 3.4** ($C$-distinguishing distribution shift). *Consider a reward-free UMDP, a pair of reward functions $R$ and $\tilde{R}$, a pair of level distributions $\Lambda^{\mathrm{Train}}, \Lambda^{\mathrm{Test}} \in \Delta(\Theta)$, a pair of ratios $\alpha, \beta \in [0, 1]$, and a constant $C > 0$. A distribution shift from $\Lambda^{\mathrm{Train}}$ to $\Lambda^{\mathrm{Test}}$ is $C$-distinguishing *if the following conditions hold.*

1. *$\Lambda^{\mathrm{Train}}$ has probability $\alpha$ on $C$-disambiguating levels and the rest on ambiguating levels.*

2. *$\Lambda^{\mathrm{Test}}$ has probability $\beta$ on $C$-disambiguating levels and the rest on ambiguating levels.*

We focus on the setting where $\alpha$ is small and $\beta$ is large. When $\alpha$ is small, the training distribution is nearly entirely ambiguous. In this setting, if $R$ is the true goal then there exists a proxy policy $\tilde{\pi} \in \Pi_\varepsilon^\star(R, \Lambda^{\mathrm{Train}}) \cap \Pi_\varepsilon^\star(\tilde{R}, \Lambda^{\mathrm{Train}}) \cap \Pi_0^\star(\tilde{R}, \Lambda^{\mathrm{Test}})$ that optimizes $\tilde{R}$ on $C$-disambiguating levels. This proxy policy makes $\tilde{R}$ a proxy goal with respect to $\Lambda^{\mathrm{Train}}$. Furthermore, when $\beta$ is large, this proxy policy performs sub-optimally with respect to the true goal $R$ after the distribution shift to $\Lambda^{\mathrm{Test}}$. If an RL system learns the proxy policy $\tilde{\pi}$ on $\Lambda^{\mathrm{Train}}$ and then is subject to a distribution shift to $\Lambda^{\mathrm{Test}}$, this is an example of goal misgeneralization.

## 4 THEORETICAL RESULTS

In Section 4.1, we prove that, while training on a fixed level distribution can lead to goal misgeneralization, regret-based prioritization of levels can prevent this. Intuitively, this is because pursuing a proxy goal on $C$-disambiguating levels generates high regret, and the adversary will play these levels, incentivizing the agent to pursue the true goal.

In Section 4.2, we give a model of the negative consequences of goal misgeneralization. In particular, we give a sufficient condition for a level to be $C$-disambiguating. This model is based on the existence of some abstract limited "resource" that, if put towards optimizing the proxy goal, would leave the true goal unfulfilled. We take inspiration from prior work on the consequences of objective misspecification (Zhuang & Hadfield-Menell, 2020). However, we adapt and generalize these arguments to the sequential decision-making context.

As a running example to illustrate these theoretical results, consider Figure 1. The agent begins with a finite number of coins, which are the limited resource. At each time step, the agent may be presented with either apples, baskets, both, or neither and can choose from actions "buy" or "move". The "buy" action acquires all the objects presented. The cost for the "buy" action is always positive. The true reward function assigns +0.5 for an apple, while the proxy assigns +0.5 to the basket.

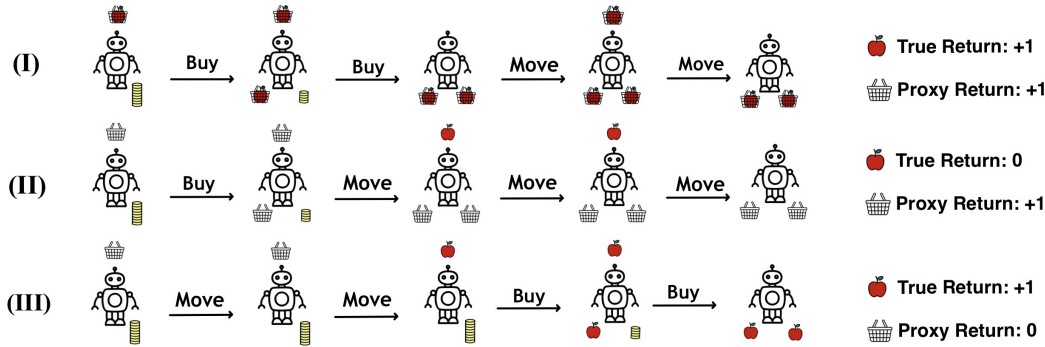

Figure 1: **Example illustrating the concepts introduced in Sections 3 and 4.** At each time step, the agent is presented with an apple, a basket, both, or neither. The true reward $R$ and the proxy $\tilde{R}$ correspond to buying apples and baskets respectively. An agent has a certain amount of resources (money), which can be allocated towards $R$ or $\tilde{R}$. (I) shows an ambiguating level. (II/III) shows a $C$-disambiguating level for $C \in (0, 1)$, in (II) the agent pursues $\tilde{R}$, while in (III) it pursues $R$.

### 4.1 MINIMAX EXPECTED REGRET MITIGATES GOAL MISGENERALIZATION

In this section, we show that under a $C$-distinguishing distribution shift, the DR objective of maximizing the expected value on the training distribution of levels permits an approximately optimal policy that performs poorly on the test distribution, as an instance of goal misgeneralization. On the other hand, we show that any policy that is approximately optimal with respect to the MMER objective is approximately optimal on the test distribution, avoiding the negative consequences of goal misgeneralization.

Rather than modeling the reasons why an optimization algorithm might prefer a proxy policy over a correctly generalizing policy, we characterize the possibility for some optimizer to select a proxy policy, regardless of inductive biases. In other words, our result does not depend on the mechanisms by which the optimization algorithm misgeneralizes.

**Theorem 4.1.** *Consider a reward-free UMDP, a pair of reward functions $R, \tilde{R}$, a pair of distributions $\Lambda^{\text{Train}}, \Lambda^{\text{Test}} \in \Delta(\Theta)$, a pair of ratios $\alpha, \beta \in [0, 1]$, and a constant $C > 0$. Let $\pi^R$ be any optimal policy w.r.t. $R$. If the distribution shift from $\Lambda^{\text{Train}}$ to $\Lambda^{\text{Test}}$ is $C$-distinguishing, then*

*(A) $\exists \pi^{\text{DR}} \in \Pi_\alpha^{\text{DR}}(R, \Lambda^{\text{Train}})$ such that $U^R(\pi^{\text{DR}}, \Lambda^{\text{Test}}) < U^R(\pi^R, \Lambda^{\text{Test}}) - \beta \cdot C$*

*(B) $\forall \pi^{\text{MMER}} \in \Pi_\alpha^{\text{MMER}}(R)$ we have $U^R(\pi^{\text{MMER}}, \Lambda^{\text{Test}}) \geq U^R(\pi^R, \Lambda^{\text{Test}}) - \alpha$*

The proof can be found in Appendix A.1.

Consider our running example. Assume the agent is trained mostly in levels like (I), where the apples are almost always sold in baskets, and then tested mostly in levels like (II/III), where the apples and baskets are mostly separated. This is a $C$-distinguishing distribution shift for $C \in (0, 1)$. Theorem 4.1(A) says it would be approximately optimal under DR for the agent to learn a policy that always buys baskets, leading to harsh consequences in the test distribution. In contrast, if training levels are prioritized to maximize regret, then Theorem 4.1(B) says that the only policies even approximately optimal will buy apples rather than baskets in the test distribution, preventing these harsh consequences.

## 4.2 GOAL MISGENERALIZATION AND ITS CONSEQUENCES

In this section we provide criteria which imply that a given level is $C$-disambiguating with respect to a specific true goal $R$ and proxy goal $\tilde{R}$. It follows that in such levels optimizing $\tilde{R}$ will lead to consequences that are negative as measured under $R$. Intuitively, these consequences can be thought of as the result of optimizing a *misspecified objective,* as has been studied previously by Zhuang & Hadfield-Menell (2020). Our argument follows a similar structure to theirs—optimizing a misspecified objective, whose performance is constrained by a limited resource, will force the agent to take that resource away from optimizing the true objective. We generalize this argument to the RL setting, and define these resources in terms of features of the local state trajectory.

An agent's *resource* could represent some physical resource explicit in the state, time in discounted MDPs, or simply the opportunity cost of taking one action instead of another. At the outset of an episode, the agent possesses some amount of this resource and decides how to allocate it throughout the episode in order to achieve a high return. We always consider *limited* resources—once consumed, a resource cannot be regained. In other words, a limited resource it is a non-increasing function along each state trajectory.

**Definition 4.2** (Limited resource). *A* limited resource *is defined as $F : S \times \mathrm{T} \to \mathbb{R}$ such that for any state trajectory $\bar{\tau} = (s_0, \dots) \in \mathrm{T}$, $\forall i < j$, it holds that $F(s_j, \bar{\tau}) \leq F(s_i, \bar{\tau})$.*

In terms of our running example, when we say that the number of coins the agent has remaining is a limited resource, we mean that it is non-increasing along each state trajectory.

While there may be many limited resources in a given level, not all limited resources cause a trade-off between two reward functions. Intuitively, the strongest trade-off would occur when a resource can be allocated to exclusively one of the reward functions. This motivates the idea of a *resource allocation* tracking the division of the resource between two reward functions across a trajectory.

**Definition 4.3** (Resource allocation). *A* resource allocation *of a resource $F$ is defined by a tuple $\langle G_R, G_{\tilde{R}}, G_{\emptyset}, f_1, f_2 \rangle$ where $G_R, G_{\tilde{R}} : S \times A \times S \to \mathbb{R}_{\geq 0}$ are functions describing how the resources allocated at a transition $S \times A \times S$ are divided amongst the two reward functions, $G_{\emptyset} : S \times A \times S \to \mathbb{R}_{\geq 0}$ tracks how much of the resource is allocated to neither reward function, and $f_1, f_2 : S \times A \times S \times \mathbb{R} \to \mathbb{R}$, are conversion functions, which describe how spent resources correspond to increased value for $R$ and $\tilde{R}$ respectively.*

*This tuple must satisfy the following conditions for all trajectories $\tau = (s_0, a_0, s_1, \dots)$:*

- *$f_1, f_2$ are monotonically non-decreasing in their last input arguments*

- *$G_R(s_t, a_t, s_{t+1}) + G_{\tilde{R}}(s_t, a_t, s_{t+1}) + G_{\emptyset}(s_t, a_t, s_{t+1}) = F(s_t, \bar{\tau}) - F(s_{t+1}, \bar{\tau})$*

- *$\sum_{t=0}^{|\bar{\tau}|-1} f_1(s_t, a_t, s_{t+1}, G_R(s_t, a_t, s_{t+1})) = U^R(\tau)$*

- *$\sum_{t=0}^{|\bar{\tau}|-1} f_2(s_t, a_t, s_{t+1}, G_{\tilde{R}}(s_t, a_t, s_{t+1})) = U^{\tilde{R}}(\tau)$*

For example, in Figure 1(II) the resource is the number of coins remaining, and in the first two time steps the resource is allocated only to the proxy reward of buying baskets, with a conversion rate of 4 coins per basket. In Figure 1(III), the resources are instead allocated to the true reward of buying apples at a rate of 4 coins per apple. In Figure 1(I), where the agent can only ever buy baskets and

apples simultaneously, there are many valid resource allocations. For instance, every buy action could allocate 2 coins to the true reward of apples and 2 coins to the proxy reward of baskets, each at an exchange rate of 0.5 reward per 2 coins.

There is a tension between maximizing $R$ and $\tilde{R}$ when there is a limited resource which is *critical* for good performance with respect to $R$ but is *marginally useful* to an agent pursuing $\tilde{R}$.

**Definition 4.4** ($C$-critical resource). *Consider a reward-free UMDP, a level $\theta \in \Theta$, a pair of reward functions $R, \tilde{R}$, a constant $C > 0$, a limited resource $F$, and a resource allocation $\langle G_R, G_{\tilde{R}}, G_\emptyset, f_1, f_2 \rangle$. $F$ is $C$-critical w.r.t. $R$ if*

$$\forall \pi, \tau \sim \pi \ s.t. \ \sum_{t=0}^{|\bar{\tau}|-1} G_R(s_t, a_t, s_{t+1}) = 0, \exists \pi' \ s.t. \ U^R(\pi', \theta) > U^R(\pi, \theta) + C$$

**Definition 4.5** (Marginally useful resource). *Consider a reward-free UMDP, a level $\theta \in \Theta$, a pair of reward functions $R, \tilde{R}$, a limited resource $F$, and a resource allocation $\langle G_R, G_{\tilde{R}}, G_\emptyset, f_1, f_2 \rangle$. $F$ is marginally useful w.r.t. $\tilde{R}$ if*

$$\forall \pi, \tau \sim \pi \ s.t. \ \sum_{t=0}^{|\bar{\tau}|-1} G_{\tilde{R}}(s_t, a_t, s_{t+1}) < F(s_0, \bar{\tau}) - F(s_{|\tau|}, \bar{\tau}), \exists \pi' \in \Pi \ s.t. \ U^{\tilde{R}}(\pi, \theta) < U^{\tilde{R}}(\pi', \theta)$$

For instance, in Figure 1(II/III), coins are needed to buy apples and do well by $R$, but are also useful to buy baskets and maximize $\tilde{R}$.

Finally, we show that if there is a resource that is critical to $R$ and marginally useful to $\tilde{R}$, then optimizing $\tilde{R}$ motivates the agent to completely exhaust the resource in pursuit of $\tilde{R}$, resulting in low reward according to $R$. It follows that the level is $C$-disambiguating in the sense of Section 3.

**Theorem 4.6.** *Consider a reward-free UMDP, a level $\theta \in \Theta$, a pair of reward functions $R, \tilde{R}$, a constant $C > 0$, a limited resource $F$, and a resource allocation $\langle G_R, G_{\tilde{R}}, G_\emptyset, f_1, f_2 \rangle$. If $F$ is $C$-critical w.r.t $R$ and marginally useful w.r.t. $\tilde{R}$, then for any policies $\pi^R, \pi^{\tilde{R}}$ optimal w.r.t. $R, \tilde{R}$ respectively,*

$$U^R(\pi^{\tilde{R}}, \theta) < U^R(\pi^R, \theta) - C$$

*In other words, $\theta$ is a $C$-disambiguating level.*

The proof can be found in Appendix A.2.

Returning to the running example, if the agent is pursuing baskets, as long as there is a way to spend coins to buy more of them, even for a significant cost, then coins are marginally useful for $\tilde{R}$. Thus the agent would want to spend all the coins on buying baskets. However, as long as there are many apples available to buy that do not come with those baskets, the coins are critical for $R$—the agent will not have any coins left to spend on the apples, resulting in low reward under $R$.

## 5 EXPERIMENTS

Regret-based UED methods have been motivated by their potential to improve sample efficiency and robustness. We have shown that, in theory, the MMER objective is also well-suited to mitigating goal misgeneralization in the face of a $C$-distinguishing distribution shift. In this section, we validate that existing UED methods are empirically capable of mitigating goal misgeneralization, showing that their level-generating adversaries are able to locate the region of the level space containing disambiguating levels, recognize when the policy has high regret in those levels, and amplify the weak training signal provided by these levels. We call this the *amplification effect*.

We construct two custom procedurally-generated grid-world environments that exhibit a $C$-distinguishing distribution shift. For each environment we construct a distribution of ambiguating levels $\Lambda_1$, a distribution of (primarily) disambiguating levels $\Lambda_2$, and from these a training distribution $\Lambda_\alpha^{\text{Train}} = (1 - \alpha)\Lambda_1 + \alpha\Lambda_2$ where $\alpha \in [0, 1]$ is a *mixture weight* controlling the proportion of disambiguating levels. We select training levels with DR in addition to a range of UED methods described in Section 5.1. We describe the environment constructions in Sections 5.2 and 5.3.

For all training runs we use a network architecture based on that of IMPALA (Espeholt et al., 2018) with a dense feed-forward layer replacing the LSTM block (following Langosco et al., 2022). We truncate rollouts at a large finite time horizon and perform policy updates with PPO (Schulman et al., 2017) and GAE (Schulman et al., 2015). We document all hyperparameters in Table 1.

## 5.1 Unsupervised environment design methods

We train policies with DR and also with the following two existing regret-based UED methods.

1. *PLR$^\perp$* (*robust prioritized level replay;* Jiang et al., 2022): Explores the level space by sampling levels from an underlying level distribution, estimating the regret of the current policy on each level, and keeping high-regret levels in a finite *level buffer*. The policy is trained on levels sampled from the buffer rather than from the underlying distribution.

2. *ACCEL* (*adversarially compounding complexity by editing levels;* Parker-Holder et al., 2023): Extends PLR$^\perp$ by, in addition to occasionally sampling new levels from an underlying level distribution, also exploring the level space in the neighborhood of the levels already in the buffer using a *mutation operation* $\mu : \Theta \to \Delta(\Theta)$ to generate similar levels. As with PLR$^\perp$, we keep high-regret levels in a level buffer used to train the policy.

We quantify the amplification effect by tracking the average frequency at which disambiguating levels are sampled from the adversary's level buffer throughout training. This gives an estimate of the average probability the adversary assigns to disambiguating levels over training. We compare this to the proportion of disambiguating levels in the underlying training distribution (the mixture weight $\alpha$).

**Regret estimation**    Both PLR$^\perp$ and ACCEL involve estimating regret. Several estimators were previously proposed by Jiang et al. (2022), but we found that they were outperformed by a simple and flexible estimator which we call the *max-latest* regret estimator, defined as

$$\hat{\mathcal{G}}^R_{\text{max-latest}}(\pi; \theta) = \hat{U}^R_{\max}(\theta) - \hat{U}^R_{\text{latest}}(\pi; \theta) \tag{3}$$

where $\hat{U}^R_{\max}(\theta)$ is the highest empirical return ever achieved for level $\theta$ throughout training (during rollouts collected with the current policy or any previous policy since the start of training), and $\hat{U}^R_{\text{latest}}(\pi; \theta)$ is the empirical average return achieved on $\theta$ in the latest batch of rollouts with the current policy $\pi$ only. We use this regret estimator for both PLR$^\perp$ and ACCEL in most of our experiments. In Appendix E, we compare to an estimator that replaces $\hat{U}^R_{\max}(\theta)$ with an oracle.

**Mutation operations**    ACCEL additionally requires specifying a mutation operation that makes $n$ various environment-specific edits to the level (e.g. moving walls or changing the agent's starting position; $n$ is a hyperparameter). We consider three distinct classes of mutation operations that differ in how they affect the balance between ambiguating and disambiguating levels, as follows.

1. *Constant goal mutation operation* (ACCEL$_c$): We make $n - 1$ random edits that do not affect goal ambiguity, followed by one edit that transforms the level into a disambiguating version of the level with probability $\alpha$ or an ambiguating version with probability $1 - \alpha$ (irrespective of whether the input level is disambiguating or ambiguating). Applying this operation to any distribution of levels results in a distribution with the same proportion of disambiguating and ambiguating levels as the training distribution.

2. *Identity goal mutation operation* (ACCEL$_{id}$): We make $n$ random edits that do not affect the goal ambiguity of the level, and no other edits. Applying this operation to any distribution of levels results in a distribution with the same proportion of disambiguating and ambiguating levels as the input distribution.

3. *Binomial goal mutation operation* (ACCEL$_{bin}$): We make a fixed number of edits $n$ where each edit is independently chosen to be either a random edit that does not affect goal ambiguity (with probability $1 - \frac{1}{n}$) or otherwise (with probability $\frac{1}{n}$) an edit that transforms the level into a disambiguating version of the level with probability $\alpha$ or an ambiguating version with probability $1 - \alpha$. This has essentially the same effect as ACCEL$_c$, except in the event that no goal ambiguity edits are sampled (with probability $(1 - \frac{1}{n})^n$) in which case it has the same effect as ACCEL$_{id}$.

Thus we experiment with a total of five methods: DR, PLR$^\perp$, ACCEL$_c$, ACCEL$_{id}$, and ACCEL$_{bin}$.

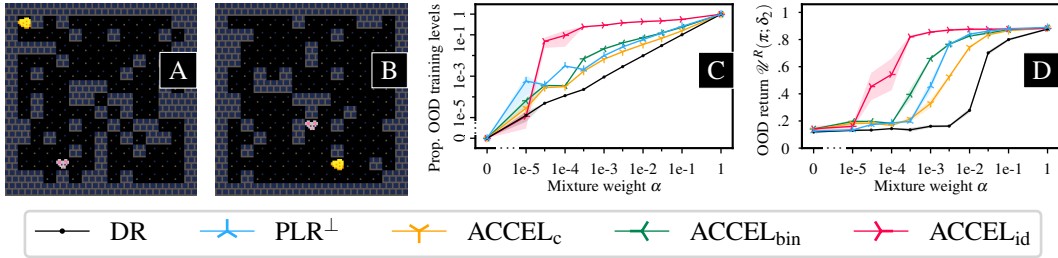

Figure 2: **Cheese in the corner.** We construct training distributions with both ambiguating levels (e.g., A) and disambiguating levels (e.g., B). We vary the mixture weight $\alpha$ (the proportion of disambiguating levels in the training distribution). We report both (C) the average proportion of disambiguating levels sampled during training and (D) the generalization performance on a fixed batch of disambiguating levels after training on approx. 250 million environment steps. We plot means over three seeds, shading shows standard error (individual runs in Appendix B.1). Except for the vertical axis in (D) we use clipped log scales with values below a given threshold labeled 0.

## 5.2 ENVIRONMENT 1: CHEESE IN THE CORNER

This environment is inspired by the Maze 1 environment from Langosco et al. (2022). The agent navigates a maze as a mouse looking for cheese. Observations are Boolean grids with one channel for the maze layout and one each for the mouse and cheese positions. The true goal assigns $+1$ reward when the mouse reaches the cheese, while the proxy goal assigns $+1$ reward the first time the mouse reaches the top left corner. Given these goals, levels with the cheese in the top left corner are ambiguating and levels with the cheese away from the corner are disambiguating.

We procedurally generate levels by randomly placing walls and the mouse. For ambiguating levels we place the cheese in the top left corner (e.g. Figure 2(A)). We generate disambiguating levels by randomly placing the cheese anywhere (e.g. Figure 2(B)). Our mutation operations delete or insert a number of walls and sometimes randomize the position of the mouse, and may toggle the level's goal ambiguity by moving the cheese into or out of the corner, as described in Section 5.1.

Figure 2(D) shows that DR is susceptible to goal misgeneralization until the mixture weight $\alpha$ is between 3e-2 and 1e-1 (that is, the training distribution has 3–10% of its mass on disambiguating levels). In contrast, all of the UED methods exhibit the amplification effect (C) and correct goal misgeneralization (D) at a much lower $\alpha$. The best performing UED algorithm, $\text{ACCEL}_{\text{id}}$, fully corrects goal misgeneralization at a very low $\alpha = \text{3e-4}$ (0.03%). Note that some of the levels in the disambiguating test distributions are unsolvable, hence why none of the methods achieve perfect return.

In Figure 3, we design an experiment to investigate the robustness of the agents trained via the methods con-

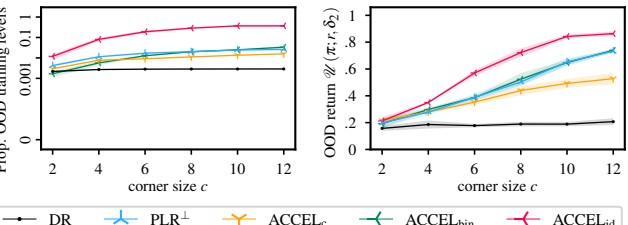

Figure 3: **Robustness experiments for cheese in the corner.** We train on a distribution comprised of ambiguating levels (cheese spawns in the corner) and disambiguating levels where the cheese spawns within a restricted corner region of size $c$. The mixture weight $\alpha$ is set to 3e-3 (other values in Appendix C.1). Runs are averaged across 3 seeds, shading shows standard error. *Left:* Proportion of disambiguating levels sampled during training. *Right:* Return on disambiguating levels where the cheese spawns anywhere in the maze.

sidered. We train our agent with the disambiguating mixture weight $\alpha = \text{3e-3}$ (we also show other values in Appendix C.1). However, we do not train on 'fully disambiguating' levels. The training distribution comprised of ambiguating levels and 'restricted' disambiguating levels where the cheese spawns in a top-left corner region of size $c$ times $c$. The return is evaluated on fully disambiguating levels, where the cheese spawns anywhere in the maze. Figure 3(Left) shows all of the UED methods exhibit the amplification effect. Figure 3(Right) shows all of the UED methods are able to make use of the subtler training signal, while DR achieves very low return across all corner sizes $c$.

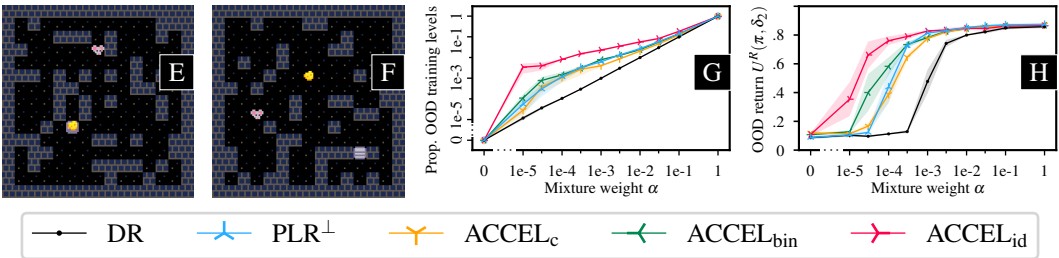

Figure 4: **Cheese on a dish.** We train with both ambiguating levels (e.g., E) and disambiguating levels (e.g., F). We vary the mixture weight $\alpha$. We report both (G) the average proportion of disambiguating levels sampled during training and (H) the generalization performance on a fixed batch of disambiguating levels after training for approx. 500 million environment steps. Means over three or more runs, shading shows standard error (individual runs in Appendix B.2). Except for the vertical axis in (H) we use clipped log scales with values below a given threshold labeled 0.

## 5.3 ENVIRONMENT 2: CHEESE ON A DISH

This environment is inspired by the Maze 2 environment from Langosco et al. (2022). This time the mouse navigates a maze containing both cheese and a secondary object—a dish. The true goal assigns $+1$ reward for reaching the cheese, while the proxy goal assigns $+1$ reward for reaching the dish. Episodes terminate when the mouse hits either object (or after a fixed time horizon). Levels with the cheese and dish co-located are ambiguating, and levels with the cheese and dish separated are disambiguating. The observations are Boolean grids with six additional channels redundantly coding the dish position (breaking symmetry to elicit a clearer case of goal misgeneralization).

We procedurally generate levels by randomly placing walls, the mouse, and the dish. For ambiguating levels, we place the cheese on the dish (e.g. Figure 4(E)). For disambiguating levels we randomly place the cheese and the dish independently (e.g. Figure 4(F)). Our mutation operations delete or insert a number of walls and sometimes randomize the position of the mouse, and may toggle the level's goal ambiguity by moving the cheese onto or away from the dish, as described in Section 5.1.

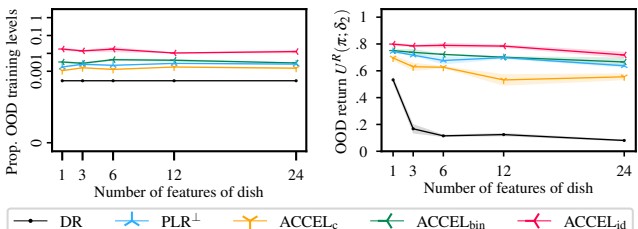

Figure 5: **Robustness experiments for cheese on a dish.** We use vary the number of channels (features) encoding the dish position. We fix $\alpha = $ 3e-4 (see also Appendix C.2). *Left:* Proportion of disambiguating levels sampled during training. *Right:* Return on disambiguating levels.

Figure 4(H) shows that DR is susceptible to goal misgeneralization until the mixture weight $\alpha$ is around 3e-3 (that is, the training distribution has around $0.3\%$ of its mass on disambiguating levels). In contrast, all UED methods exhibit a strong amplification effect (G), and reduce goal misgeneralization (H) at significantly lower $\alpha$. ACCEL$_{id}$ corrects goal misgeneralization at a notably low $\alpha = $ 1e-4 (that is, $0.01\%$).

In Figure 5, we vary the inductive bias through the number of channels coding the dish position at a fixed mixture weight $\alpha = $ 3e-4 (other values in Appendix C.2). The UED methods are significantly more robust than DR. Additional channels significantly increase DR's susceptibility to goal misgeneralization, while UED methods retain comparably similar performance. The amplification effect appears essentially constant. We hypothesize that the number of channels does not stop the adversary from noticing high-regret disambiguating levels, though it may affect how the policy responds.

## 6 RELATED WORK

**Unsupervised environment design (UED).** Our work shows that UED methods are powerful tools to improve *inner alignment* of reinforcement learning agents. While it is well-known that UED

methods improve capabilities generalization and general robustness of RL agents, this work is the first demonstration of their power for improving alignment of RL agents as well. UED was formalized by Dennis et al. (2020). Following works have since proposed additional UED algorithms, like PLR$^{\perp}$ and ACCEL used in this work (Jiang et al., 2022; Parker-Holder et al., 2023).

**Goal misgeneralization.** Alignment of reinforcement learning agents is a very challenging problem. A distinction can be made between *outer alignment* and *inner alignment* (Ngo et al., 2023). Inner misalignment occurs when an agent fails to robustly pursue the preferences of the principal despite *correct* specification of the reward function (Hubinger et al., 2019). Goal misgeneralization can be considered a type of inner alignment failure. Inner alignment failures can also arise due to lack of adversarial robustness (Lu et al., 2023). Compared to outer alignment, very few works exist that are explicitly focused on assuring inner alignment of RL agents. Specific to goal misgeneralization, Langosco et al. (2022) and Shah et al. (2022) demonstrated the occurrence of goal misgeneralization in various setups, but do not propose any methods to mitigate it. Starace (2023) approaches goal misgeneralization as 'task-underspecification' and conditions decision transformer style models on natural language descriptions of goals instead of scalar reward values. This is a complementary approach aimed at influencing the inductive biases in RL to favor the true goal in more circumstances.

**Underspecification.** Goal misgeneralization in some sense arises from underspecification present in the training setup (Shah et al., 2022). When a machine learning problem is underspecified, multiple models may exist that behave similarly on in-distribution data but behave in qualitatively different ways on out-of-distribution data (Teney et al., 2022). Prior works have discussed how underspecification can lead to erroneous evaluation of reinforcement learning (Jayawardana et al., 2022), and machine learning models in general (D'Amour et al., 2022). Underspecification underlying goal misgeneralization is also related to the identifiability of reward functions. Goal misgeneralization occurs when there exist proxies that highly correlate with the true reward function on the training domain, thus, posing a challenge for the learning agent to correctly identify the *right* reward function. A reward function is generally not identifiable from behavioral data within a single environment (Ng et al., 2000; Skalse et al., 2023; Schlaginhaufen & Kamgarpour, 2023). However, in multi-environment setups, it is sometimes possible to uniquely identify the reward function (Amin et al., 2017; Cao et al., 2021; Büning et al., 2022; Rolland et al., 2022).

# 7 CONCLUSIONS

We studied goal misgeneralization arising from a nearly ambiguous training distribution followed by a $C$-distinguishing distribution shift. In this setting, we have contributed a combination of theoretical and empirical results characterizing the ability of the minimax expected regret (MMER) objective and regret-based unsupervised environment design (UED) methods to mitigate goal misgeneralization. Our results also highlight the pitfalls of training with fixed level distributions.

We have provided a formal framework of the potential negative consequences of goal misgeneralization in sequential decision-making. Using this framework, we have proven the first theoretical guarantees of the effectiveness of different training methods in avoiding these negative consequences in the setting of $C$-distinguishing distribution shifts.

We have also empirically established that existing regret-based UED methods are capable of amplifying weak training signals favoring correct goal generalization via the amplification effect. These results signal UED's potential as a defense against goal misgeneralization—the amplification effect is in a position to become increasingly powerful as UED methods improve since more successful UED methods should be even more capable of detecting and amplifying rare, high-regret levels. Moreover, while prioritization-based UED methods like PLR$^{\perp}$ are confined to the support of the training distribution, ACCEL is only limited by the span of its mutation operations, and more powerful UED methods could in principle surface disambiguating levels from anywhere in the level space.

Future work must address the setting of fully ambiguous training distributions and distribution shifts beyond the reach of UED algorithms, in which case we again face the challenge of understanding the internal dynamics of our learning algorithms and our learned policies. We are hopeful that our work will instigate further research on the problem of goal misgeneralization, which remains a critical, open problem in alignment and safe generalization of reinforcement learning agents.

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

# A    THEORETICAL RESULTS AND PROOFS

## A.1    PROOF OF THE DR AND MMER THEOREM (THEOREM 4.1)

**Theorem 4.1.** *Consider a reward-free UMDP, a pair of reward functions $R, \tilde{R}$, a pair of distributions $\Lambda^{\mathrm{Train}}, \Lambda^{\mathrm{Test}} \in \Delta(\Theta)$, a pair of ratios $\alpha, \beta \in [0, 1]$, and a constant $C > 0$. Let $\pi^R$ be any optimal policy w.r.t. $R$. If the distribution shift from $\Lambda^{\mathrm{Train}}$ to $\Lambda^{\mathrm{Test}}$ is $C$-distinguishing, then*

*(A)* $\exists \pi^{\mathrm{DR}} \in \Pi_\alpha^{\mathrm{DR}}(R, \Lambda^{\mathrm{Train}})$ *such that* $U^R(\pi^{\mathrm{DR}}, \Lambda^{\mathrm{Test}}) < U^R(\pi^R, \Lambda^{\mathrm{Test}}) - \beta \cdot C$

*(B)* $\forall \pi^{\mathrm{MMER}} \in \Pi_\alpha^{\mathrm{MMER}}(R)$ *we have* $U^R(\pi^{\mathrm{MMER}}, \Lambda^{\mathrm{Test}}) \geq U^R(\pi^R, \Lambda^{\mathrm{Test}}) - \alpha$

*Proof.* We will prove each part separately.

**Part A.** For simplicity we assume $\Theta$ is discrete. Denote $\bar{\Theta}$ as the subset of levels in $\Lambda^{\mathrm{Train}}$ such that $\theta \in \bar{\theta}$ if and only if $\theta$ is $C$-disambiguating. First, consider the utility function optimized by DR. Given the probability distribution $\Lambda^{\mathrm{Train}}$, we have (defining $r^K$ as the expected return of a policy on a given level w.r.t. to the reward function $K$),

$$U_{\mathrm{DR}}^K(\pi, \Lambda^{\mathrm{Train}}) = \sum_{\theta \in \mathrm{supp}(\Lambda^{\mathrm{Train}})} \mathbb{P}_{\Lambda^{\mathrm{Train}}}(\theta) \cdot r^K(\pi, \theta)$$

Let $\pi^{\mathrm{DR}}$ be any perfectly optimal policy w.r.t. $\tilde{R}$ in all levels.

We first need to prove that $\pi^{\mathrm{DR}} \in \Pi_\alpha^{\mathrm{DR}}(R, \Lambda^{\mathrm{Train}}) = \Pi_\alpha^\star(R, \Lambda^{\mathrm{Train}})$. We can see that

$$U_{\mathrm{DR}}^R(\pi^{\mathrm{DR}}, \Lambda^{\mathrm{Train}}) = \sum_{\theta_i \in (\mathrm{supp}(\Lambda^{\mathrm{Train}}) \setminus \bar{\Theta})} \mathbb{P}_{\Lambda^{\mathrm{Train}}}(\theta_i) \cdot r^R(\pi^{\tilde{R}}, \theta_i) + \sum_{\theta_j \in \bar{\Theta}} \mathbb{P}_{\Lambda^{\mathrm{Train}}}(\theta_j) \cdot r^R(\pi^{\tilde{R}}, \theta_j)$$

$$\geq \sum_{\theta_i \in (\mathrm{supp}(\Lambda^{\mathrm{Train}}) \setminus \bar{\Theta})} \mathbb{P}_{\Lambda^{\mathrm{Train}}}(\theta_i) \cdot r^R(\pi^{\tilde{R}}, \theta_i)$$

$$= \sum_{\theta_i \in (\mathrm{supp}(\Lambda^{\mathrm{Train}}) \setminus \bar{\Theta})} \mathbb{P}_{\Lambda^{\mathrm{Train}}}(\theta_i) \cdot r^R(\pi^R, \theta_i)$$

where the third equality holds by Definition 3.2 and Definition 3.4. Thus, due to returns being normalized this satisfies the definition of $\Pi_\alpha^\star(R, \Lambda^{\mathrm{Train}})$, and it follows that $\pi^{\mathrm{DR}} \in \Pi_\varepsilon^\star(R, \Lambda^{\mathrm{Train}})$.

Now, we are left to show that $U^R(\pi^R, \Lambda^{\mathrm{Test}}) - U^R(\pi^{\mathrm{DR}}, \Lambda^{\mathrm{Test}}) > \beta \cdot C$.

Rewrite the utility on test of $\pi^{\mathrm{DR}}$ as

$$U_{\mathrm{DR}}^R(\pi^{\mathrm{DR}}, \Lambda^{\mathrm{Test}}) = \sum_{\theta_i \in (\mathrm{supp}(\Lambda^{\mathrm{Test}}) \setminus \bar{\Theta})} \mathbb{P}_{\Lambda^{\mathrm{Test}}}(\theta_i) \cdot r^R(\pi^{\tilde{R}}, \theta_i) + \sum_{\theta_j \in \bar{\Theta}} \mathbb{P}_{\Lambda^{\mathrm{Test}}}(\theta_j) \cdot r^R(\pi^{\tilde{R}}, \theta_j)$$

By Definition 3.3 it must be the case that on all of the levels in $\bar{\Theta}$, $U^R(\pi^R, \theta) - U^R(\pi^{\mathrm{DR}}, \theta) > C$ holds. So, since this set $\bar{\Theta}$ has probability $\mathbb{P}_{\Lambda^{\mathrm{Test}}} = \beta$, the set of levels $\theta_i \in (\mathrm{supp}(\Lambda^{\mathrm{Test}}) \setminus \bar{\Theta})$ must have probability $1 - \beta$.

Thus, since we are guaranteed of a difference of at least $C$ in $\bar{\Theta}$, it holds that

$$U^R(\pi^R, \Lambda^{\mathrm{Test}}) - U^R(\pi^{\mathrm{DR}}, \Lambda^{\mathrm{Test}}) > \sum_{\theta_j \in \bar{\Theta}} \mathbb{P}_{\Lambda^{\mathrm{Test}}}(\theta_j) \cdot C = \beta \cdot C$$

This concludes the proof of (A).

**Part B.** Suppose for purposes of contradiction that

$$U^R(\pi^R, \Lambda^{\mathrm{Test}}) - U^R(\pi^{\mathrm{MMER}}, \Lambda^{\mathrm{Test}}) > \alpha$$

Then by Equation (1)

$$\mathbb{E}_{\theta \sim \Lambda^{\mathrm{MER}}}\left[ \mathcal{G}^R(\pi^{\mathrm{MMER}}; \theta) \right] \geq \mathbb{E}_{\theta \sim \Lambda^{\mathrm{Test}}}\left[ \mathcal{G}^R(\pi^{\mathrm{MMER}}; \theta) \right] > \alpha$$

But by assumption, $\pi^{\mathrm{MMER}} \in \Pi_\alpha^\star(R, \Lambda^{\mathrm{MER}})$, so

$$\mathbb{E}_{\theta \sim \Lambda^{\mathrm{MER}}}\left[ \mathcal{G}^R(\pi^{\mathrm{MMER}}; \theta) \right] \leq \alpha$$

which is a contradiction. This concludes the proof for (B).    $\square$

## A.2 Proof of the Resource Theorem (Theorem 4.6)

**Theorem 4.6.** *Consider a reward-free UMDP, a level $\theta \in \Theta$, a pair of reward functions $R, \tilde{R}$, a constant $C > 0$, a limited resource $F$, and a resource allocation $\langle G_R, G_{\tilde{R}}, G_\emptyset, f_1, f_2 \rangle$. If $F$ is $C$-critical w.r.t $R$ and marginally useful w.r.t. $\tilde{R}$, then for any policies $\pi^R, \pi^{\tilde{R}}$ optimal w.r.t. $R, \tilde{R}$ respectively,*

$$U^R(\pi^{\tilde{R}}, \theta) < U^R(\pi^R, \theta) - C$$

*In other words, $\theta$ is a $C$-disambiguating level.*

*Proof.* By definition, $U^R(\pi^R, \theta)$ is the optimal utility achievable under $R$. We want to show that $U^R(\pi^{\tilde{R}}, \theta)$ is low. By definition, it must be that since the resource is marginally useful for $\tilde{R}$ and $\pi^{\tilde{R}}$ is optimal according to $\pi^{\tilde{R}}$, any trajectory $\tau = (s_0, a_0, \ldots)$ with positive probability under $\pi^{\tilde{R}}$ must devote all resources towards $\tilde{R}$:

$$\sum_{t=0}^{|\bar{\tau}|-1} G_{\tilde{R}}(s_t, a_t, s_{t+1}) = F(s_0, \bar{\tau}) - F(s_{|\bar{\tau}|}, \bar{\tau})$$

But if that is the case, because of the conditions on the resource allocation, it must hold that

$$\sum_{t=0}^{|\bar{\tau}|-1} G_R(s_t, a_t, s_{t+1}) = 0$$

Given the fact that the resource is critical for $R$, $\exists$ some $\pi$ such that $U^R(\pi) > U^R(\pi^{\tilde{R}}) + C$. But because $\pi^R$ is optimal w.r.t $R$, $U^R(\pi^R) \geq U^R(\pi)$. It follows that

$$U^R(\pi^{\tilde{R}}) < U^R(\pi^R) - C$$

Now since this holds for all optimal policies $\pi^{\tilde{R}} \in \Pi_0^\star(\tilde{R}, \theta)$ it follows that

$$\Pi_0^\star(\tilde{R}, \theta) \cap \Pi_C^\star(R, \theta) = \emptyset$$

This concludes the proof. $\qquad\square$

# B  INDIVIDUAL RUNS

## B.1  ENVIRONMENT 1: CHEESE IN THE CORNER

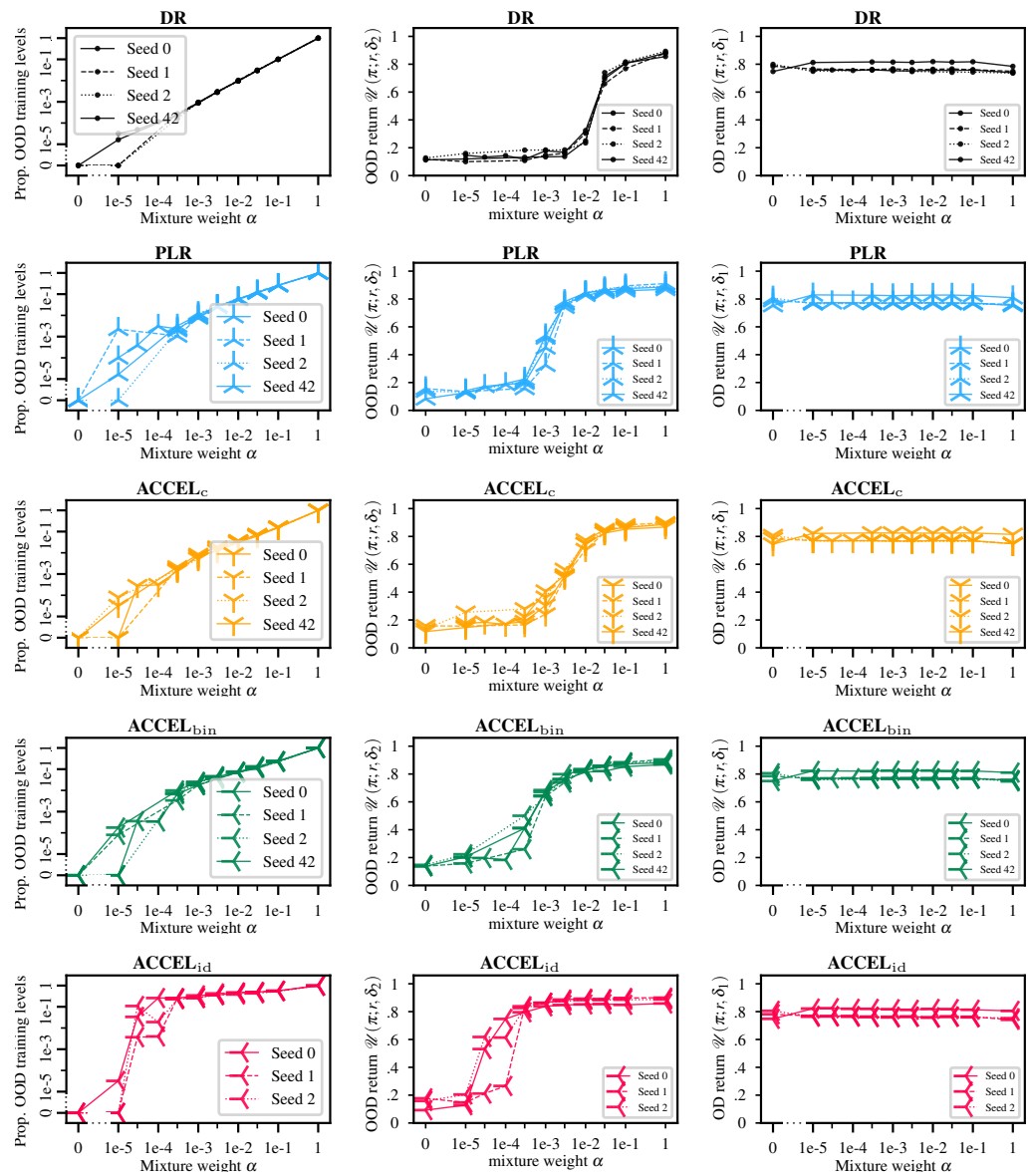

Figure 6: **Individual runs for cheese in the corner experiments in Figure 2.** Each row corresponds to a training algorithm, displaying the proportion of disambiguating levels during training (left), the average return on disambiguating levels (OOD levels) at the end of training (center), and the the average return on ambiguating levels (OD levels) at the end of training (right).

## B.2  ENVIRONMENT 2: CHEESE ON A DISH

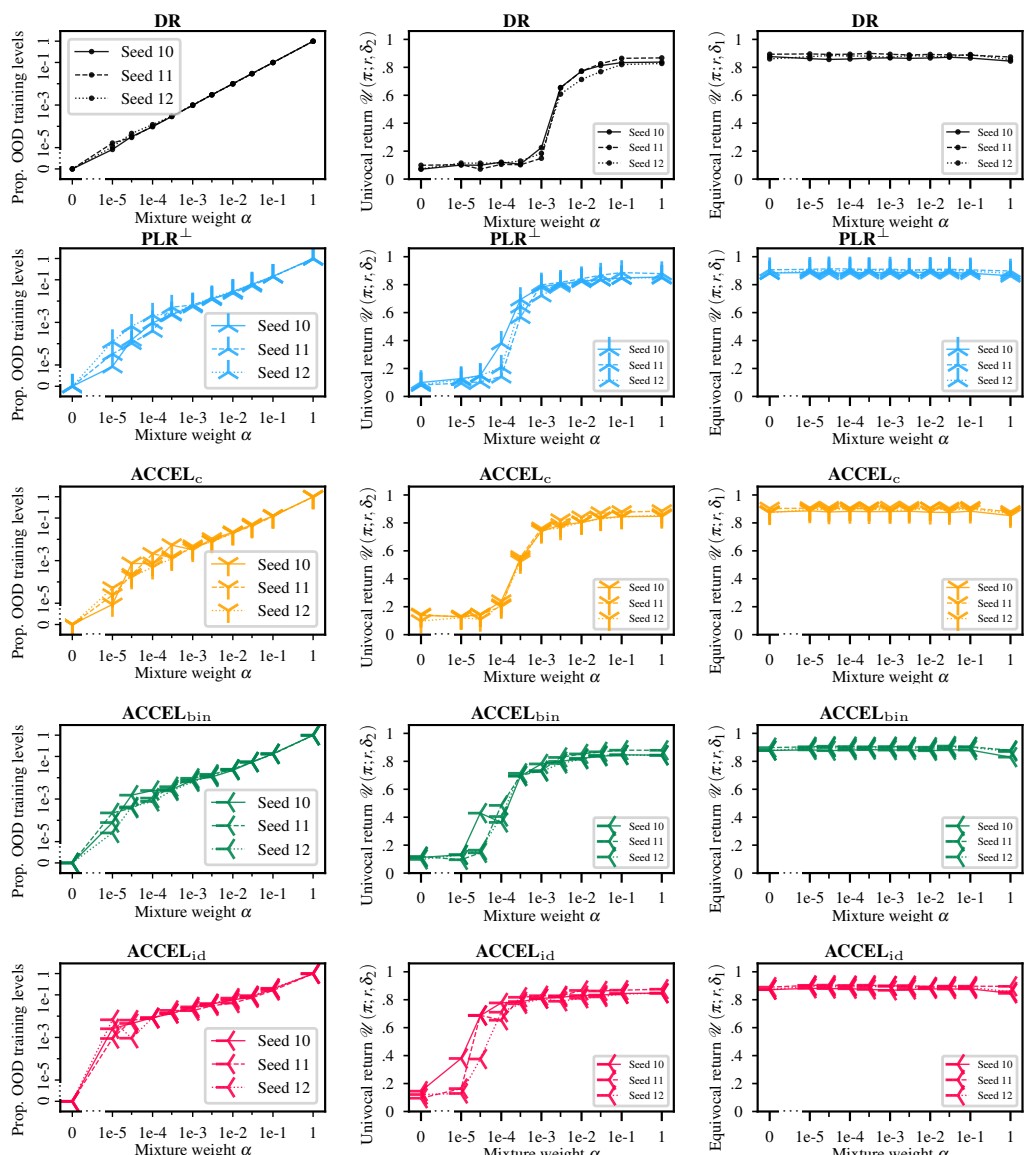

Figure 7: **Individual runs for cheese on a dish experiments in Figure 4.** Each row corresponds to a training algorithm, displaying the proportion of disambiguating levels during training (left), the average return on disambiguating levels at the end of training (center), and the average return on ambiguating levels at the end of training (right). Note: The axis labels are from an old version of the terminology. Equivocal levels = ambiguating levels and univocal levels = disambiguating levels.

## C  ADDITIONAL ROBUSTNESS EXPERIMENTS

### C.1  ENVIRONMENT 1: CHEESE IN THE CORNER

We provide additional experiments similar to Figure 3 with different mixture weights.

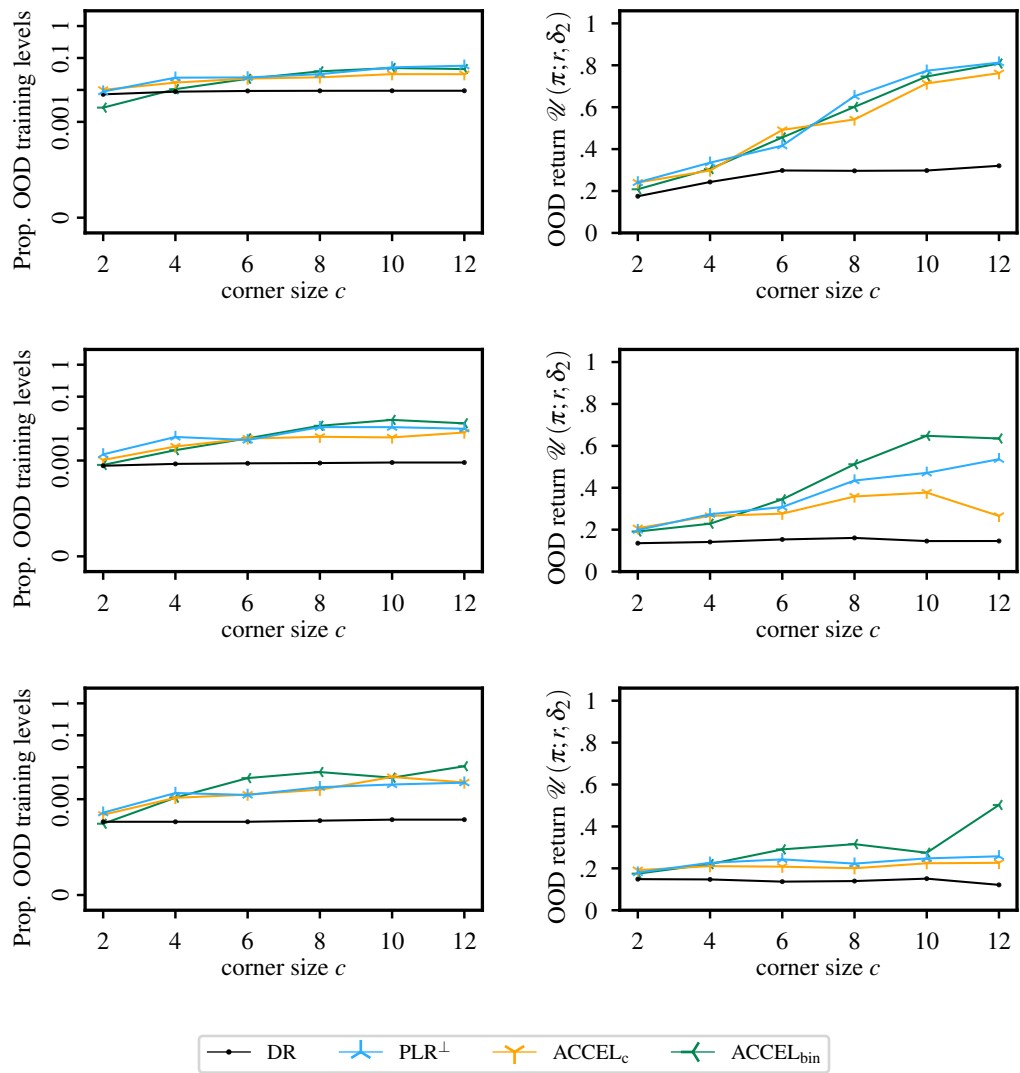

Figure 8: **Additional robustness experiment, same setting as Figure 3.** *Left column:* Average proportion of disambiguating levels during training. *Right column:* Return on disambiguating levels. *First row:* $\alpha$ set to 1e-2. *Second row:* $\alpha$ set to 1e-3. *Third row:* $\alpha$ set to 3e-4.

## C.2 ENVIRONMENT 2: CHEESE ON A DISH

We provide additional experiments similar to Figure 5 with different mixture weights.

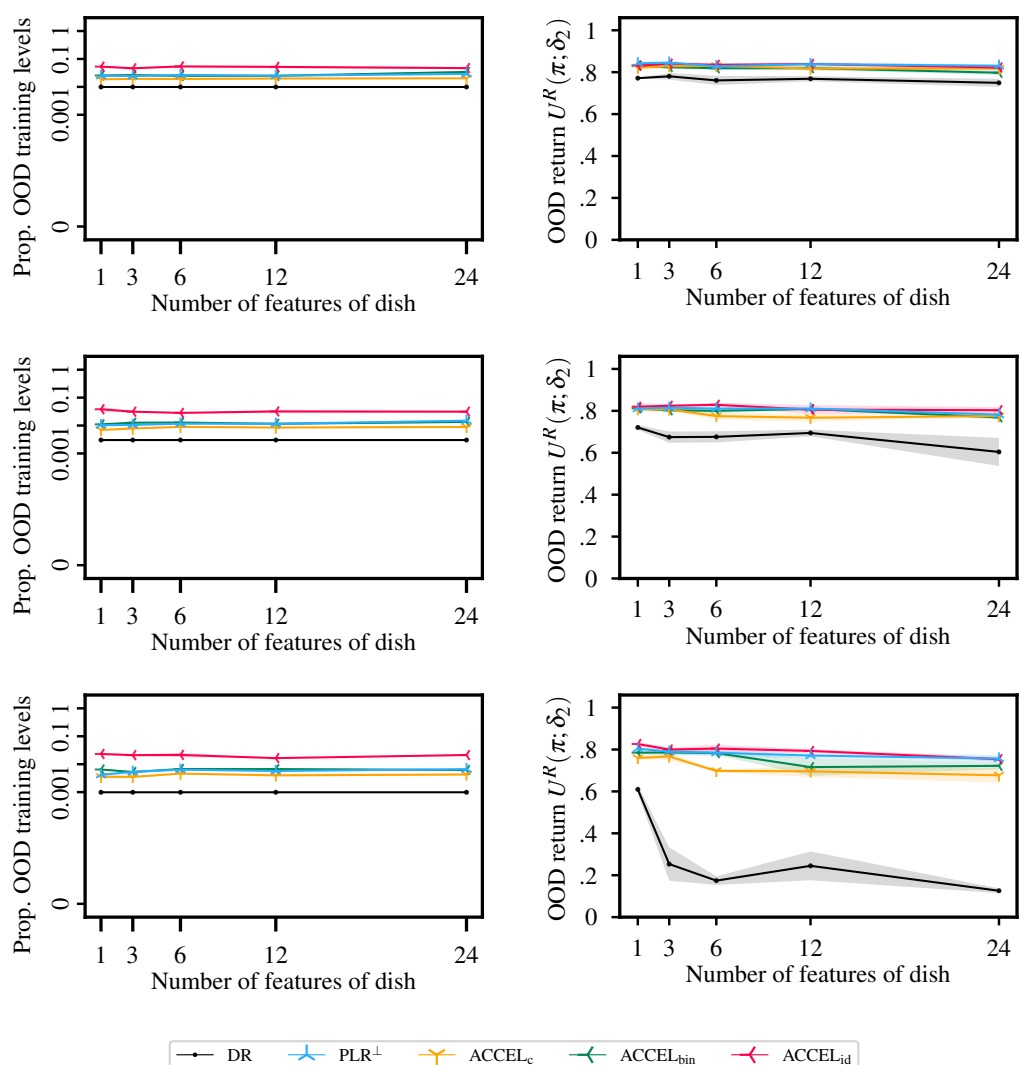

Figure 9: **Additional robustness experiment, same setting as Figure 5.** *Right column:* Return on disambiguating levels. *Left column:* Average proportion of disambiguating levels during training. *First row:* $\alpha$ set to 1e-2. *Second row:* $\alpha$ set to 3e-3. *Third row:* $\alpha$ set to 1e-3.

## D HYPERPARAMETERS AND IMPLEMENTATION DETAILS

We report all the relevant hyperparameters for our experiments in Table 1. The values have been chosen according to previous work, together with an exploratory analysis which considered individually the following subsets of parameters:

- learning rate in $\{1e\text{-}5, 5e\text{-}5, 1e\text{-}4, 5e\text{-}4\}$
- discount factor $\gamma$ in $\{0.999, 0.99, 0.995\}$
- PPO entropy coefficient in $\{0.0001, 0.001, 0.01\}$
- replay rate for ACCEL in $\{0.5, 0.7, 0.9\}$
- replay rate for PLR in $\{0.33, 0.5\}$
- regret estimators $\{$ PVL, MaxMC, max-latest $\}$
- PLR staleness coefficient in $\{0.1, 0.3\}$
- PLR temperature in $\{0.1, 0.3\}$
- ACCEL number of mutations in $\{2, 4, 6, 12, 16, 32, 64\}$

When only the first value is compiled for a row, it indicates that all the methods used the same hyperparameter.

Table 1: Hyperparameters used for training each method.

| Parameter | DR | PLR$^\perp$ | ACCEL$_c$ | ACCEL$_{id}$ | ACCEL$_{bin}$ |
|---|---|---|---|---|---|
| *PPO* | | | | | |
| $\gamma$ | 0.999 | | | | |
| $\lambda_{GAE}$ | 0.95 | | | | |
| PPO rollout length | 128 | | | | |
| PPO epochs | 5 | | | | |
| PPO minibatches per epoch | 4 | | | | |
| PPO clip range | 0.1 | | | | |
| PPO # parallel environments | 256 | | | | |
| Adam learning rate | 5e-5 | | | | |
| PPO max gradient norm | 0.5 | | | | |
| PPO value clipping | yes | | | | |
| PPO critic coefficient | 0.5 | | | | |
| PPO entropy coefficient | 1e-3 | | | | |
| | | | | | |
| *UED* | | | | | |
| Replay rate, $p$ | – | 0.33 | 0.5 | 0.5 | 0.5 |
| Buffer size, $K$ | – | 4096 | 4096 | 4096 | 4096 |
| Regret estimator | – | max-latest | max-latest | max-latest | max-latest |
| Prioritization | – | rank | rank | rank | rank |
| Temperature, $\beta$ | – | 0.1 | 0.1 | 0.1 | 0.1 |
| Staleness coefficient | – | 0.1 | 0.1 | 0.1 | 0.1 |
| | | | | | |
| *ACCEL* | | | | | |
| Number of mutations per step | – | – | 12 | 12 | 12 |

# E ADDITIONAL EXPERIMENTS WITH ORACLE MAX RETURN

In Section 5, we showed that even with the simple max-latest estimator (equation (3)), existing UED methods are capable of mitigating the negative effects of goal misgeneralization in some cases. We argue that our theoretical results assuming a strong adversary show that this benefit will increase as more sophisticated and powerful UED methods are developed. As an example of this dynamic, we ran a variant of the experiment in Section 5.2 (Figure 2) using a modified *oracle-latest* regret estimator using knowledge of the true maximum return for each level,

$$\hat{\mathcal{G}}^R_{\text{oracle-latest}}(\pi;\theta) = \max_{\pi'} U^R(\pi';\theta) - \hat{U}^R_{\text{latest}}(\pi;\theta) \tag{4}$$

where $\hat{U}^R_{\text{latest}}(\pi;\theta)$ is, as in Section 5.1, the empirical average return achieved by the current policy. The maximum return for a level, $\max_{\pi'} U^R(\pi';\theta)$, is computationally intractable to obtain in a general environment. However, in cheese in the corner, we can easily compute the optimal return as $\gamma^d$ where $d$ is the length of the shortest path from the mouse's spawn position to the cheese position.

In Figure 10, we plot the generalization performance of the four UED methods alongside their variant based on the oracle-latest regret estimator. The results show the variants with more powerful regret estimators significantly outperform their counterparts. In particular, Figure 10 (bottom left/right) show that the oracle-based UED methods are able to mostly mitigate goal misgeneralization even for mixture weight $\alpha = $ 1e-5.

These results show there is ample room for improvement for regret estimation. A plausible mechanism for the under-performance of the max-latest estimator is that it does not notice that a level has high true regret if the current policy never produces a trajectory containing the cheese. In this situation, the UED methods may still form a curriculum and mitigate goal misgeneralization, but not as effectively. Future work in UED may reveal more effective strategies for regret estimation, such as using a separate policy network to estimate the maximum return along the lines of Dennis et al. (2020).

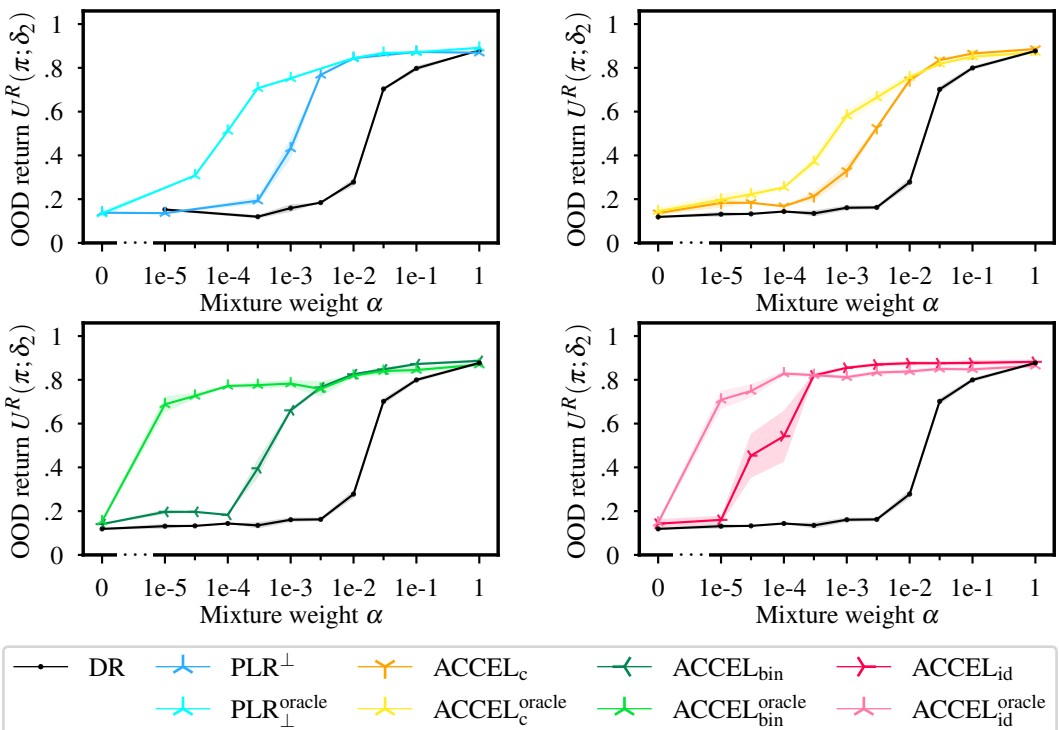

Figure 10: **Cheese in the corner, estimated max return vs. oracle max return.** Generalization performance on a fixed batch of disambiguating levels after training for approx. 250M time steps, using DR (black baseline) or a UED algorithm using either the max-latest estimator (equation (3), dark line) or the oracle-latest estimator (equation (4), light line). Shading: standard error, 3 seeds.

## F    ADDITIONAL EXPERIMENTS WITH KEYS AND CHESTS ENVIRONMENT

In addition to the environments considered in the main text, this appendix considers experiments in a third environment inspired by the Keys and Chests environment studied in Langosco et al. (2022).

In this environment, once again, a mouse navigates a maze grid-world. This maze contains both keys and chests. The mouse collects keys and uses collected keys to open chests. The true goal assigns $+1$ reward for each open chest. Episodes terminate when the mouse collects all available (true) reward, or after a fixed maximum episode length. Levels are procedurally generated with randomized wall, key, chest, and mouse positions. The observations are Boolean grids with one channel respectively coding the position(s) of the walls, mouse, keys, and chests, and a fifth channel used to code the presence of keys that have been collected but not yet used to open chests.

Consider a distribution of *sparse-key* levels with a small number of keys and a large number of chests. One proxy goal assigns $+1$ reward for collecting each key and a small positive reward for opening each chest. An example proxy policy collects all available keys and then subsequently opens as many chests as keys collected. Sparse-key levels are ambiguating (at least approximately, see below). *Dense-key* levels, with a large number of keys and a small number of chests, are disambiguating—pursuing the proxy goal incentivizes collecting all of available keys before opening any chests.

While still a grid-world, this environment is more complex than those studied in Section 5. Firstly, rather than receiving reward for carrying out a single navigation task, the policy has to achieve a compound task of navigating first to a key and then to a chest. In fact, since there are multiple keys and chests, the policy has to solve multiple of these tasks concurrently. Secondly, as noted, sparse-key levels are only *approximately* ambiguating. Optimizing the proxy goal by collecting all keys before opening any chests is approximately optimal according to the true goal, since the reward from delayed chest opening is slightly discounted. Thirdly, the nature of goal ambiguity in sparse-key levels is richer. Collecting keys is subject to delayed reinforcement, and ambiguity arises as to whether the key collection was merely instrumentally valuable or was intrinsically valuable.

Langosco et al. (2022) showed that in a similar ProcGen environment (Cobbe et al., 2020) that a policy trained on sparse-key levels with DR can misgeneralize, collecting keys in dense-key levels. We extend this finding to the case where the training distribution also contains a small proportion of dense-key levels. We study UED methods $\text{PLR}_\perp^{\text{heuristic}}$ and $\text{ACCEL}^{\text{heuristic}}$ which estimate the maximum return available on a level with a simple heuristic ($+1$ for each reachable chest if there is a corresponding reachable key, not accounting for discounting). For $\text{ACCEL}^{\text{heuristic}}$, the mutation operation randomly toggles a subset of walls or moves a subset of keys, chests, and the mouse. We use the hyperparameters from Table 1 excepting a smaller number of mutations per step (6) and a larger PPO entropy coefficient (1e-2) to encourage more exploration for this complex task. Figure 11 shows the results. We see that DR is susceptible to goal misgeneralization until $\alpha = 1\text{e-}1$ (10%), whereas $\text{PLR}^\perp$ successfully mitigates goal misgeneralization at $\alpha = 3\text{e-}2$ (3%) and ACCEL mitigates goal misgeneralization at all positive mixture weights tested (including as low as $\alpha = 3\text{e-}4$).

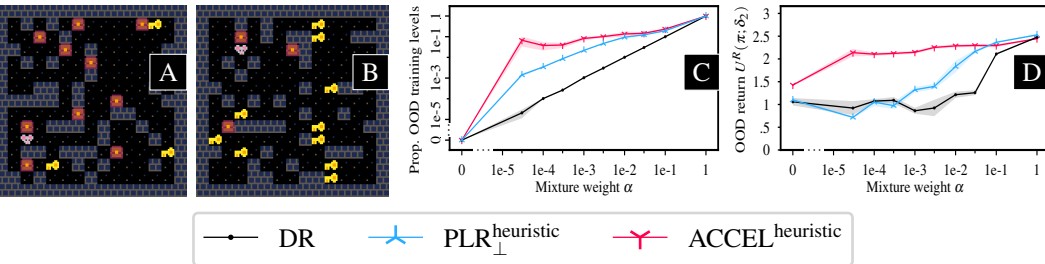

Figure 11: **Keys and chests.** We construct training distributions with both (approximately) ambiguating levels (e.g. A) and disambiguating levels (e.g. B). We vary the mixture weight $\alpha$ (the proportion of disambiguating levels in the training distribution). We report both (C) the average proportion of disambiguating levels sampled during training and (D) the generalization performance on a fixed batch of disambiguating levels after training on approximately 400 million environment steps. We plot the mean value over three seeds with the shaded region marking the standard error. Except for the vertical axis in (D) we use clipped log scales with values below a given threshold labeled 0.

