# OpenReview forum: "Mitigating Goal Misgeneralization via Minimax Regret"
_ICLR.cc/2025/Conference — Submitted to ICLR 2025_

### Official Review · Reviewer_T3jU · 2024-10-16

**Soundness:** 2
**Presentation:** 2
**Contribution:** 2
**Rating:** 3
**Confidence:** 4

**Summary:**

In this paper, the authors analyse the problem of goal misegeneralization, i.e., the problem that arises when our learning agent optimizes a different reward function (goal, named proxy reward) instead of the true one. While the true rewards induce similar optimal policies during training, good policies under the proxy reward may be arbitrarily bad under the true reward during testing. The authors analyse a sort of constrained problem, in which there are resources (with a budget) associated to reward functions. They show that good policies on environments with large regret can limit goal misgeneralization. They also find that, empirically, unsupervised environment design can mitigate the effects of goal misgeneralization.

**Strengths:**

- The considered problem of goal misgeneralization is interesting and important for the RL community

**Weaknesses:**

- Basically the author consider a constrained RL problem, but do not mention constrained RL nowhere, not even in the related works.
- The paper is difficult to follow because there are some counter-intuitive parts, like the fact that the resources/budget constraints are associated to reward functions instead of state-actions. I think that this formulation is not a model that might be applied in many applications. Instead, I think that remodelling the problem using a cost function as a function of the state/action would be much more meaningful and a much more powerful model.

Moreover, there are many typos:
- 190: missing "is"
- 191: two "the"
- Definition 4.1: clarifying that $s_i,s_j$ belong to the trajectory
- 240: "allocationof"
- 087,088: the return of a trajectory should not depend on $\theta$
- 241,248: it is wrong notationally to write $\sum\limits_{s_t\in\tau}$, but you have to loop also over $a_t,s_{t+1}$ if you want to use them
- ...

**Questions:**

- why not modelling the problem with a state with two components, i.e., using one component for the resource?   Or, alternatively, why not modelling the problem using both a reward function and a cost function, with the latter for the budget/resource constraints?
- definition 4.2: what is G? It has no reward associated with it.
- I do not understand the meaning of allocating resources to rewards. If the resource is a function of the state (according to your definition 4.1), and since a reward is a function of the state too, why should the resource be allocated to rewards in different manners?

---

> ### Author Response · Authors · 2024-11-22
>
> Thank you for your review, questions, and suggestions. We think we can address your concerns as follows.
>
> **Clarification: We are not doing constrained RL.** We respectfully disagree that we consider a constrained RL problem. We apologize for creating this impression, which we think we caused by not being sufficiently clear about the aims of our model of the consequences of goal misgeneralization.
>
> To clarify, we exclusively consider a classical unconstrained RL problem without any constraint/cost function. However, in one of our theory sections (section 4.1 in the submitted paper, section 4.2 in the revision), we give a sufficient condition for consequences of goal misgeneralization based on the existence of an abstract limited resource that two reward functions compete for. This finite resource superficially resembles a constraint as in constrained RL. However, there is a crucial difference in aims between this part of our work and constrained RL:
>
> * We are aiming to model a classical RL environment as having an *implicit* constraint, and thereby study the consequences for a classical RL algorithm solving that environment, given no knowledge of the existence of the implicit constraint.
> * In contrast, constrained RL starts with an augmented environment with an explicit constraint and considers the problem of solving that environment subject to not violating the constraint, and constrained RL algorithms are allowed to make use of explicit knowledge of the constraint.
>
> In our latest revision, we have clarified this role of the resource framework, as part of a significant refactoring of the theory sections described in the top-level comment. In particular, we made the main result about the performance of MMER vs. DR independent of the resource competition framework, so that hopefully it will now be clearer that this result applies to classical RL.
>
> We hope that this explanation and the revisions clarify the differences in aims between our resource framework and constrained RL. Has this addressed your concern?
>
> **Questions about resource framework:** Keeping in mind the distinction between the aims of our resource framework and those of constrained RL, we offer the following answers to your questions:
>
> * **Q1: Why don’t we model the state with a separate component for the resource?** Our aim is to model *implicit* constraints given any environment. We don’t want to assume that the resource is explicitly represented in the state. The resource may be a feature of the state, but we don’t want to restrict ourselves to this setting. Thus we allow the resource to be any non-increasing function of the states along each state trajectory, not just a particular component.
>
> * **Q2: What is G in the resource allocation definition?** Good question. We don’t assume that spent resources are always allocated to one or the other reward function, they may be allocated to neither reward function. We use $G$ to track this. We agree this name is unclear, in the latest revision we have renamed it to $G\_{\\emptyset}$ and explicitly described its purpose.
>
> * **Q3: What is the meaning of allocating resources to rewards?** Note that the finite resource is a function along a state trajectory. The resource allocation function tracks when the consumption of the resource coincides with the receipt of rewards from one, both, or neither reward function.
>
>   In the running example from the paper, if the policy takes a ‘buy’ action in a state with a basket, then we model that by saying some of the resource is allocated to the proxy reward function,  whereas if the policy takes a ‘buy’ action in a state with an apple, then we say the resource has been allocated towards the true reward function. If the policy takes a ‘buy’ action in a state with both, then some of the resource is allocated to both reward functions. Does this answer your question?
>
> **On the clarity of the theory:** We are committed to improving the clarity of the paper and appreciate your suggestions. As mentioned, we have significantly refactored the theory sections in our latest revision. Regarding your suggestion to use an explicit cost function, could you please explain your suggestion in more detail, taking into account our clarification of the goals of this part of our work? While we aren’t studying a constrained RL problem, we are definitely interested in opportunities for insights from constrained RL to lead to a better abstract model of the consequences of goal misgeneralization in classical RL.
>
> **Typos:** Thank you for pointing these out. We have addressed them all in the revision.
>
> **Summary:** We thank you once again for your questions and suggestions. We hope that in light of the above clarification of the aims of our resource framework we have resolved your main concern regarding the relationship to constrained RL in particular, and we invite you to consider raising your score if so. Otherwise, please let us know if you have any remaining concerns or questions.

---

> > ### Comment · Reviewer_T3jU · 2024-11-23
> >
> > I would like to thank the authors for having answered to my questions and comments. However, I believe that the presentation of the contents in the paper is too difficult to read and understand, and, as such, does not permit to fully convey the ideas of the paper. For this reason, I do not think that this paper, in these conditions, meets the standards of ICLR, and I will recommend reject.
> > I would like to suggest the authors to improve the presentation of the paper if they want to submit the paper again to other venues.

---

### Official Review · Reviewer_DDwf · 2024-10-24

**Soundness:** 2
**Presentation:** 2
**Contribution:** 2
**Rating:** 5
**Confidence:** 5

**Summary:**

The paper addresses the important problem of goal misgeneralization in reinforcement learning (RL) and provides formalisms using level ambiguity. The authors propose that level sampling in Unsupervised Environment Design (UED) based on minimax expected regret (MMER) can mitigate the effects of goal misgeneralization by amplifying the training signal from disambiguating levels. The paper theoretically proves that MMER-based methods are more robust to goal misgeneralization than standard domain randomization (DR) methods using the resource-allocation setting. Empirical results in a single grid-world domain validate these claims, and show that regret-based Unsupervised Environment Design (UED) methods significantly outperform DR in avoiding goal misgeneralization.

**Strengths:**

1. The paper tackles the important problem of goal misgeneralization, which is highly relevant in reinforcement learning, particularly in safety-critical applications. The application of minimax regret to mitigate goal misgeneralization is a novel approach.

2. The paper formalizes the problem of goal misgeneralization using level ambiguity and introduces a framework based on minimax regret. Based on my knowledge on previous works regarding goal misgeneralization in deep RL, the distinction between disambiguating and ambiguating levels is a novel contribution. It is also very intuitive for understanding the goal misgeneralization problem.

3. The paper has strong theoretical rigor and provides a extensive proofs that minimax expected regret policies mitigate goal misgeneralization.

**Weaknesses:**

**1. There is a lack of experiments**

Gridworld environments is fine since previous works in deep RL studying "goal misgeneralization" have mainly experimented using those [1][2]. But I am wondering why there isn't more experiments in different domains such as "Keys and Chest" from [1] and especially in domains with denser rewards such as "Tree Gridworld" from [2]. Testing in only one domain makes it difficult to assess whether the observed improvements are due to the MMER strategy itself or are influenced by domain-specific factors, such as the reward structure (e.g., sparse vs. dense rewards). Broader testing across different domains is needed to establish the generality of the MMER approach.

**2. Results visualizations are very confusing and lack explanations**

The plots are really not intuitive and readers would stuggle to interpret them. In Figures 2C and 2G, the y-axis shows "prop of OOD training levels" and x-axis shows "mixture weight". Correct me if I'm wrong, but there isn't a line in the paper explaining what the mixture weight $\alpha$ is, i.e. whether it is portion of ambiguating or disambiguating levels. Same for prop of OOD training axis. What is the difference between the two axis? As of current, how to interpret both figures is unclear to me.

In lines 412-413, it is stated that "DR is susceptible to goal misgeneralization until there is 3-10% of training levels that are disambiguating" but in line 420, "$\text{ACCEL}_\text{id}$ ... at a very low $\alpha$ = 0.03%". Is the percentage in the latter statement intended? Why does $\alpha$ have a max of 1 in the plots but then it is mentioned in percentages in the explanations.

**3. Empirical evidence is not conclusive**

Based on the results, specifically Figure 2D and Figure 4G, it appears that MMER is not primary factor mitigating goal misgeneralization. The performance of most of the regret-based UED approaches closely tracks that of DR, only showing divergence at a mixture weight of around 1e-3 in "Cheese in the corner". This is with the exception of $\text{ACCEL}_\text{id}$. It appears that the mutation method for environment generation has more impact than the minimax regret level sampling, which diminishes the significance of MMER. Also, the use of only 3-4 seeds per experiment is below the typical standard in deep RL research, especially given the inherent variability in RL performance.

**4. Overcomplicated and confusing theoretical section**

A large part of the paper is devoted towards defining and proving using the "resource-allocation" setting which is arguably a restrictive setting of sequential decision making problems. This proof is also unnecessarily complicated for what it's trying to show. Wouldn't the key point be to bring across that MMER strategy prioritizes levels where the policy has high regret, which would likely be the disambiguating levels (since they are OOD)? It would be much better deferred to the appendix and having more of the paper spent on empirical results (e.g. more domains) and providing clearer interpretations of the effects of MMER.


**5. It is unclear whether only MMER strategy is viable for solving goal misgeneralization or just one of many.**

The MMER strategy is potentially just one of several approaches for mitigating goal misgeneralization. The significance of using MMER to address goal misgeneralization is not established as the paper lacks comparision against other alternatives for level sampling. For instance, it is possible that the maximin reward strategy could similarly target the goal misgeneralization problem. Additionally, novelty-based approaches, where more diverse levels are prioritized (which would potentially contain more disambiguating goals), might be equally effective. Furthermore, the challenge of goal misgeneralization may not solely stem from how levels are sampled, but also from the learning algorithm itself. Improvements in the "inner-loop" mechanisms, such as better exploration through intrinsic curiosity, might be far better at addressing the problem. While the paper demonstrates the impact of MMER, it does not fully establish why this method is superior to other possibly simpler or more intuitive strategies. If several alternative approaches could potentially mitigate goal misgeneralization, the necessity of using MMER in particular is not so important.



[1] Langosco Di Langosco, Jack Koch, Lee D Sharkey, Jacob Pfau, and David Krueger. Goal misgeneralization in deep reinforcement learning. In Proceedings of the 39th International Conference on Machine Learning, volume 162 of Proceedings of Machine Learning Research, pp. 12004–12019. PMLR, July 2022.

[2] Rohin Shah, Vikrant Varma, Ramana Kumar, Mary Phuong, Victoria Krakovna, Jonathan Uesato, and Zac Kenton. Goal misgeneralization: Why correct specifications aren’t enough for correct goals, 2022.

**Questions:**

**1. Why does corner size matter in "Cheese in the corner"?**

Could you explain the corner size $c$ and the plot with x-axis as $c$ is shown in the main body? How is $c$ related to the ambiguity of the level?

**2. Why is there a need for log scale for the vertical axis in figures 2D and 4G?**

Isn't both Gridworld experiments are already in the scale of reward being 0-1, where 1 represents mouse successfully getting the cheese in every run?

**3. Why is there no robustness result for "Cheese on a dish" in the appendix?**

How does ACCEL and PLR scale with different feature numbers?

**4. Possible reason why max-latest regret estimator has the best result**

From my understanding of minimax-regret UED approaches as in PLR and ACCEL, positive GAE (generalized advantage estimation) has typically been better performing. However, you mentioned that the max-latest regret estimator which does not rely on the policy's value estimate, but rather keeps track of the running max achieved performance on a level, achieved the best result. It could potentially be due to the fact that ambiguating levels causes the policy to overestimate the value in disambiguating levels. Intuitively, within disambiguating levels (which form a low proportion of the training distribution), the agent always overestimate the value of states/trajectory leading to the common location of the cheese, but in reality gets zero reward. This lead to consistently negative GAE values, which are zeroed out under positive GAE, effectively hiding regret information. Therefore using the max attained reward which does not rely on value estimate allows the regret to be in the positive range. If that is true, the approach’s effectiveness may depend heavily on learning rate (LR) and entropy regularization in PPO rather than on the MMER approach alone. A low LR combined with higher entropy (encouraging exploration) would lead the agent to "accidentally" find the goal in disambiguating levels more often, thereby reinforcing positive regret signals under the max-regret estimate. This may suggest that MMER is not entirely reliable in isolation. It might be good to discuss and experiment regarding this in order to establish MMER's significance.

**5. Questions in the "Weaknesses" section**

---

> ### Author Response · Authors · 2024-11-22
> **Reply (part 1 of 4)**
>
> We are deeply thankful for your detailed review and your thoughtful feedback and suggestions. We have taken most of your suggestions on board and incorporated improvements and additional experiments into our latest revision, and we think this has improved the paper. In particular, your comments about the theoretical sections have prompted us to thoroughly reevaluate our presentation of the theory, and this has made the paper dramatically clearer and more cohesive in our assessment.
>
> We respond in detail to each of your concerns (W) and questions (Q) below. As mentioned, we have taken most of them on board, however there are a small number of your concerns that we think we can make a strong case against, which we have attempted to do. We apologize for the lengthy message, which discusses each point in detail. Thanks again for your thoughtful review and valuable suggestions, and we welcome further discussion.
>
> ---
>
> **W1: On lack of experiments.** We think we have strengthened the paper with **additional experiments in the ‘keys and chests’ environment.** A new appendix F contains experiments in a JAX implementation of an environment like ‘Keys and Chests’ from Langosco et al. (2022). A full description of the environment, experiment, and results is in appendix F in the latest revision, and summarized in the [top-level comment 1/2](https://openreview.net/forum?id=po67tkP0Jx&noteId=6ICGCI2do0).
>
> Regarding the ‘Tree Gridworld’ environment from Shah et al. (2022), note that this environment demonstrates goal misgeneralization in a continual learning setting in a long-horizon task where there is a distribution shift of the states encountered within a single (long) episode. We think addressing goal misgeneralization in this setting is a priority for future work, but don’t consider it in scope.
>
> **W2: Clarity of empirical results.** We agree with these points and we think we have addressed them, improving the clarity of the results section, as follows:
>
> * **Mixture weight $\\alpha$:** We inconsistently referred by the three names (1) ‘$\\alpha$’ (in the opening of section 5 when the variable is defined), (2) ‘mixture weight $\\alpha$’ (in the x axis label), and (3) ‘proportion of disambiguating levels in the training distribution’ (figure captions) to one concept, the probability of sampling a disambiguating level from the fixed training distribution.
>
>   To rectify this, in the latest revision we consistently use the term ‘mixture weight $\\alpha$’ (or occasionally just ‘$\\alpha$’ where it is clear from context) for the first variable. We introduce alpha as a mixture weight controlling the proportion of disambiguating levels (line 322 in the revision). The axis labels already use this convention. We changed the invocations in the figure captions and text to the new convention.
>
> * **Proportions vs. percentages:** When referring to the value of this $\\alpha$ variable, we  followed two distinct conventions, namely using (1) the raw mixture weights between 0 and 1 (in the figures and occasionally in the text) and also (2) the corresponding proportions as percentages between 0% and 100% (mainly in the text). This explains why the x axes only go up to 1—this is the maximum raw proportion. We believe we always correctly used the percentage symbol, but we didn’t consistently use percentages vs. raw proportions in a principled way.
>
>   To rectify this, in the latest revision we *always* use raw proportions in both figures and text, and in some cases where it aids in interpreting the raw proportions we *additionally* express the value as a percentage (but in some cases we give only the raw proportion).
>
> * **Mixture weight vs. prop. OOD training levels:** We refer by (1) ‘proportion of OOD training levels’ (figures y axis) and also (2) ‘the average proportion of disambiguating levels sampled during training’ to a distinct variable from the mixture weight $\\alpha$. Namely, this refers to the empirical proportion of levels sampled throughout training that are disambiguating. Note that for DR, this variable closely tracks $\\alpha$ (hence the straight line), but for the UED methods it is instead an attempt to quantify the amplification effect by measuring the weight that the UED adversary gives to disambiguating levels. Our hypothesis, and finding, is that it will generally be higher than alpha.
>
>   In the revised paper, we have added a paragraph in section 5.1 (line 342\) that elaborates on this quantity and its purpose.
>
> Do these answers and revisions address your concern?

---

> ### Author Response · Authors · 2024-11-22
> **Reply (part 2 of 4)**
>
> **W3a: Empirical evidence is not conclusive (UED methods don’t always mitigate goal misgeneralization and mutation methods have more impact).** We disagree that this should be considered a concern. It’s true that for sufficiently low mixture weight, some UED methods are also subject to goal misgeneralization (including for ACCELid at very low mixture weight). It’s also true that the mutator plays a significant role in the extent to which ACCEL mitigates goal misgeneralization.
>
> However, we think this evidence is consistent with our claims. Our theoretical results say that UED methods with an ideal adversary always mitigate goal misgeneralization. In practice we do not have an ideal adversary, but we show that current UED methods are capable of mitigating goal misgeneralization in some cases, which is consistent with our theory. But our theory also predicts that the stronger the adversaries (the better they are able to find a high-regret level distribution) the more able to mitigate goal misgeneralization they will be. Our observations support these hypotheses:
>
> * **Current UED methods out-perform DR:** All UED methods we tried dominate DR in that they mitigate goal misgeneralization for a wider range of mixture weights than DR does.
> * **The least restricted adversary, ACCELid, is the most effective at mitigating goal misgeneralization.** ACCELid is the least restricted due to its ability to search the space of level distributions through cumulative mutation operations that can easily identify disambiguating levels and then experiment with mutations of these same levels without them spontaneously mutating back into ambiguating levels.
> * **Extra experiments with even more powerful regret estimators achieve even better performance.** Similarly, when we conducted extra experiments with a maximum return oracle (see [top-level comment 2/2](https://openreview.net/forum?id=po67tkP0Jx&noteId=E5lRQeVyqL) and appendix E in the revision), taking another step towards an ideal adversary, we found that the performance improved substantially again.
>
> The implication is that the more powerful UED methods we develop, the more successful they will become at mitigating goal misgeneralization. Does this perspective make sense? Has it addressed your concern?
>
> **W3b: Empirical evidence is not conclusive (not enough seeds).** We are happy to invest in the compute required to increase our main experiments from 3 seeds to a total of 8 seeds (approximately 500 additional training runs). We will upload a new revision when we have collected the results. We fully expect the results to be consistent with what we have now, and will keep you updated.
>
> **W4: Overcomplicated theory.** Thank you again for the feedback and suggestions. Following your suggestions we believe we have made significant improvements to the clarity of the theory in our latest revision.
>
> Please see the revised paper and the detailed summary of changes in the [top-level comment 2/2](https://openreview.net/forum?id=po67tkP0Jx&noteId=E5lRQeVyqL). As a brief summary touching on your specific points of feedback, we have taken on board your suggestion to restate the main result directly in terms of disambiguating levels rather than in terms of the resource framework. This now appears before the resource framework in section 4.1, and we have deferred all mention of resources and their associated complexity to section 4.2. We think of the resource framework as a core contribution, so we are reluctant to move it to the appendix (we think it makes more sense to put additional experiments in the appendix, since they are similar in content to what is already in the main text).
>
> Have these revisions addressed your concern about the overcomplicated presentation of the theory?

---

> ### Author Response · Authors · 2024-11-22
> **Reply (part 3 of 4)**
>
> **W5: MMER is not significant because it is not a unique approach.** We agree that MMER/UED is not a unique approach to mitigate goal misgeneralization, and we apologize if we gave a different impression. However, we disagree that this is a weakness of our work or takes away from our work’s significance.
>
> * **There is relatively little concrete prior work on mitigating goal misgeneralization.** While we agree there are many potential approaches, as far as we are aware none of the specific methods you have suggested have ever been tried, so we have no published baselines to compare to. We see developing such baselines as non-trivial work that is out of scope for this paper.
>
> * **Some alternative approaches could complement UED,** meaning studying UED is still significant. For example, we mention the complementary approach of improving inductive biases in the related work section. Your particular suggestions of improving exploration would also be complementary to UED methods, since better exploration also improves regret estimates, thereby helping the adversary as well as the policy.
>
> * **We believe MMER-based UED is a particularly theoretically principled approach** among alternative level selection-based approaches we are aware of, and we think this gives us good reason to focus on it first. Prioritizing levels using maximin return or some notion of diversity seems like it could work in some cases, but these approaches will not benefit from the strong theoretical guarantees of MMER (from our paper and prior work like [Dennis et al., (2020](https://proceedings.neurips.cc/paper/2020/hash/985e9a46e10005356bbaf194249f6856-Abstract.html), theorem 1)). For example, an adversary that minimizes level return will generally fail to identify disambiguating levels in the presence of a large number of unsolvable levels. Likewise, diversity-based level sampling will fail to pick out a small number of disambiguating levels when there are a large number of factors of level variability that are not relevant to the goal.
>
> Ultimately, we claim that **identifying and convincingly demonstrating a single method that is theoretically and practically successful in solving a problem that has not previously been solved by any method should count as a significant contribution.** Moreover, as we write in our concluding sentence, our work invites ‘further research on the problem of goal misgeneralization,’ which we intend to mean both research on both enhanced UED methods and also non-UED methods that are able to outperform our approach. Does this address your concern?

---

> ### Author Response · Authors · 2024-11-22
> **Reply (part 4 of 4)**
>
> Finally, we turn to addressing your questions:
>
> **Q1: Corner size:** Good question. In this experiment, disambiguating levels are generated with the cheese located in a small $c$ by $c$ square window starting in the top corner. For example, if $c=6$, then the cheese could spawn anywhere in roughly the top left quadrant of the maze, and for $c=12$ almost the entire maze (like the experiments in Figure 2\. Thus, the training distribution weights ambiguating levels (i.e. corner size 1), with probability $1-\\alpha$ and disambiguating levels with corner size $c$ with probability $\\alpha$. However, the agent is then tested in levels where the cheese can spawn anywhere. With a small corner size, disambiguating levels are less clearly disambiguating because the cheese and the corner are close together in the maze, so, in most cases, similar trajectories will achieve both the true goal as the proxy goal. This experiment shows that our UED methods are able to pick up on a much weaker signal than if the disambiguating levels are more obviously disambiguating, and this can still mitigate goal misgeneralization to some extent. We believe this shows a nice robustness property of UED methods in our setting.
>
> **Q2: Log scale in figures 2D and 4G.** Could you please clarify the question? There is no log scale on the vertical axis in figure 2D, or any of our axes plotting return (you are correct that return is in \[0,1\] in these environments). There is a log scale on the vertical axis in figure 4G, but it does not show return, it shows the proportion of sampled OOD levels (see W2) which is on a log scale like the mixture weight alpha.
>
> **Q3: Where are robustness results for cheese on a dish:** Sorry that this was unclear. The number of features is already plotted on the x-axis. For cheese on a dish, we had only conducted the experiment shown in Figure 5\. In contrast, for the similar experiment with cheese in the corner (figure 3), appendix C showed additional experiments at a couple of different mixture weights.
>
> It’s a good idea to also try the cheese on a dish robustness experiment at different mixture weights. We did this and added the results in a new appendix C.2.
>
> The results are as expected: As in the original experiment shown in Figure 5, in these new experiments, UED methods see only a slight drop in performance as the number of features increases, but DR experiences a more severe drop. The exception is mixture weight 1e-2, in which case DR also robustly mitigates goal misgeneralization (though to a slightly lesser extent than the UED methods, consistent with the UED literature’s findings that UED methods can lead to improved robustness in non-ambiguous environments).
>
> **Q4: On regret estimation:** Good point about positive GAE-based regret estimation hiding regret information in this setting. To our knowledge, prior work in UED also found the MaxMC estimator similarly effective compared to positive GAE (e.g., Jiang et al., 2022). As you may be aware, MaxMC is similar to max-latest (same ‘max’ estimator, but instead of estimating the current policy’s return using the latest rollout, it uses the value network). From your observation that the value network may be prone to overestimate the value of disambiguating levels it follows that MaxMC would underestimate the regret even if the policy happens to achieve the true goal. Having said all this, max-latest is also prone to underestimating regret in the more common situation that exploration is insufficient for the policy to stumble upon any true reward and realize what it is missing.
>
> Our perspective on all of this is as follows. It seems that regret estimation for UED is at a relatively immature stage, and it would be worth experimenting with better alternative regret estimation methods, especially methods that are capable of accurately estimating regret for rare disambiguating levels given a misgeneralizing policy and critic. As we discuss in our [top-level comment 2/2](https://openreview.net/forum?id=po67tkP0Jx&noteId=E5lRQeVyqL) and appendix E, our new oracle-based regret estimation experiments show there is room for improvement. We would guess that, like you suggest, factors like exploration can indeed make a big difference in the success of current UED methods in this setting (in our new keys and chests experiments we had to increase the entropy coefficient to get UED methods to work). However, given our theoretical results, we view the unreliability of current UED methods in isolation as a limitation of today’s methods, rather than a limitation of MMER-based UED as a paradigm, so we still think our overall contribution is pointing to something significant.
>
> ---
>
> **Summary:** Thank you once again for your detailed review and thoughtful suggestions. We are of course happy to continue discussing any follow-up questions. We hope we have sufficiently addressed your concerns that you might consider raising your score.

---

> ### Comment · Reviewer_DDwf · 2024-11-23
> **Response to Authors (part 1/2)**
>
> Thank you authors for the detailed rebuttal. Please find below my responses to the changes made.
>
> > We agree with these points and we think we have addressed them, improving the clarity of the results section
>
> I can't speak for other reviewers unfamiliar with UED work, but it is much clearer to me now. I would still advise the authors on revising the paper more in further iterations (acceptance or future resubmissions) to make the work more readable to the masses.
>
> > While we agree there are many potential approaches, as far as we are aware none of the specific methods you have suggested have ever been tried, so we have no published baselines to compare to. We see developing such baselines as non-trivial work that is out of scope for this paper.... We believe MMER-based UED is a particularly theoretically principled approach among alternative level selection-based approaches we are aware of, and we think this gives us good reason to focus on it first.
>
> I respectfully disagree. Many of my suggestions are simple to implement and **necessary** baselines in this field of work. First, minimax comparison stems from robust RL and it only requires the teacher's utility to be the minimum reward achieved in levels. Minimax comparison has been deeply rooted in UED literature, see PAIRED, PLR and ACCEL. Other baselines I suggested such as more exploration and intrinsic curiosity are also trivial to implement. While this necessitate more experiments to be run, given that your environments (and possible algorithms) are implemented in JAX, it should be a non-issue. Note that origin works, and many current works now still, in UED are still based on DCD, which is **significantly** slower. And these works include DR, minimax, PLR, ACCEL, and many more baselines. Adding more experiments which are necessary for substantiating MMER benefits should be something the authors are actively pursuing.
>
> > As we discuss in our top-level comment 2/2 and appendix E, our new oracle-based regret estimation experiments show there is room for improvement.
>
> I would sincerely suggest the authors to keep Occam's razor and simplicity in mind. Adding more methods of regret estimation distracts the readers from the true purpose of the paper -- showing the purpose of regret in mitigating goal misgeneralization. In my own personal advice, sticking with one basis and preferably more general estimator of regret such as PVL or MaxMC, and consistently applying it throughout the paper would significantly reduce the confusion regarding the paper and make its contributions more poignant. Afterwards, dedicate other resources to studying the isolated effects of the chosen minimax regret estimator compared to other alternatives outside of minimax regret such as Minimax, which I stressed in my previous statements.
>
> > A new appendix F contains experiments in a JAX implementation of an environment like ‘Keys and Chests’ from Langosco et al. (2022)
>
> Thank you for including additional experiments as requested. The results have made 2 things clear to me:
> 1. The solutions proposed for solving MMER is heavily reliant on *regret engineering*. The choice of using a new heuristic for regret calculation rather than the max-latest regret appears to me that different regret design is necessary for different environments. This goes back to my point, if such fine-grained information about the max reward attainable needs to be designed (e.g. in oracle and heuristic), it detracts from the generality of the MMER approach.
> 2. A lot of the empirical performance still comes from the level presentation rather than MMER (refer to my next point)

---

> ### Comment · Reviewer_DDwf · 2024-11-23
> **Response to Authors (part 2/2)**
>
> ## Suggestion for possible future experiment
>
> One potential flaw of the current experiments is that the DR agent's policy simply does not even understand what the goal is, or is not even capable of learning the proxy goal. This often happens in RL generalization where constantly changing environment dynamics confuses the policy so much such that it plateaus to almost a random policy. This is likely why ACCEL performs much better than DR and PLR; it gradually introduces more complex environment dynamics. In order to ablate the effects of minimax regret vs DR, this experiment would be much stronger. Train a DR agent purely on **only ambiguating levels** till saturation, i.e. it can always find the cheese. Afterwards, using that pretrained agent, perform more training with inclusion of disambiguating levels, but comparing DR vs MMER (ACCEL/PLR). You want to make sure that the agent learns to recognize how to get to the proxy goal first, then show that more training with DR causes the agent to stick to finding the proxy goal location, but more training with MMER causes the agent to start recognizing the true goal. I think this experiment would really lend to the credibility of your approach, but ymmv.
>
> ## Concluding comments
>
> I don't see much scope for me to increase the score. I would raise it to a 4, mainly because of the extra experiments, but ICLR does not offer that liberty. I sincerely believe that this work has huge potential, but its novelty and contributions are not strong enough at current stage for publication. I hope my suggestions would be of use to future iterations. I look forward to following this work.

---

> > ### Author Response · Authors · 2024-12-02
> > **Thanks again**
> >
> > Thank you again for your deep engagement with us during the discussion period, for your thoughtful comments, and especially for recognising the potential of our work.
> >
> > **Regarding your latest suggestion for a new experiment:** You mentioned that our results could be driven by DR and PLR failing to learn a coherent policy due to constantly changing dynamics. However, our original experiments already showed this is not the case. The rightmost column of figures 6 and 7 in the appendix show that **DR/PLR achieves near-optimal return on ambiguating levels**. This implies that all our policies, including DR, learn coherently and are capable of pursuing the proxy goal. Since you suggested that including an experiment showing these results would “really lend to the credibility of \[our\] approach”, we invite you to consider this evidence in deciding your final rating.
> >
> > **Regarding your comments on “regret engineering” and “Occam’s razor”:** We believe these concerns have been driven by us not being sufficiently clear about our paper’s core claim. Our core claim is that regret-based UED algorithms are capable of mitigating goal misgeneralization *in principle.* This claim is supported by our strong theoretical result and exemplified by our experiments.
> >
> > * Note however that, we do not claim that *current* instantiations of UED methods are sufficient to handle more complex environments than those we study. In appendix F we find that current methods are in fact not capable of mitigating goal misgeneralization, but we show that if we emulate a more powerful regret estimator, then we do mitigate goal misgeneralization. This is consistent with our core claim.
> > * Likewise in appendix E on oracle regret estimation, rather than seeing this as adding more regret estimators, we see this as more closely emulating the theory by replacing part of the estimator with a ground truth value. This showcases a limitation of regret estimation of current UED methods as you and other reviewers have correctly noted.
> >
> > We are making this core claim more explicit for future revisions or submissions.
> >
> > Thanks again for your consideration.

---

> ### Comment · Reviewer_DDwf · 2024-12-03
> **Thank you authors**
>
> Thank you authors for replying to my comments.
>
> ### Experiment suggestion
> Thank you for pointing our DR's ability to find the goal in disambiguating levels.  However, I still believe that implementing the suggested experiment—where all algorithms begin with a shared policy pretrained to identify the proxy goal and demonstrating how PLR/ACCEL can guide the policy toward aligning with the intended goal would provide stronger empirical support for your claims. This would be a valuable addition to future iterations of your work.
>
> ### Clarity of work
> As noted by other reviewers and yourself, the manuscript could benefit from greater clarity in articulating its core claim. Adding many confusing elements such as the resource theorem or multiple regret estimators certainly do not help your case. I must also remind that *"strong"* is an overstatement of the theoretical results you have established. In fact, most of the relevant theoretical contributions are summarized in Appendix A.1. The current work still lack theoretical and empirical comparisons against Minimax, an established UED baseline, in terms of addressing goal misgeneralization. I still retain the belief that additional comparisons would be important if you want to strengthen the case for your paper. However, it appears my advice will likely fall on deaf ears since you have insisted that it is "out of scope" and have instead opted to concentrate on other experiments, such as using the "oracle regret estimator," during the discussion period.
>
> In recognition of the promising direction of your research, I have raised my score to 5. I still think the research have many notable gaps and is not of publication standards at the current moment. I encourage the reviewers to be more receptive to reviewers' feedback. Good luck.

---

> > ### Author Response · Authors · 2024-12-03
> >
> > We are sorry to have given the impression that we are not receptive to reviewer feedback. We reiterate that we feel your review in particular was very valuable in guiding our revision. Given limited capacity for revisions and additional experiments during the discussion period, we chose to prioritize refactoring the theory and exploring an additional environment, which were among your many suggestions. We also included oracle regret experiments we had already prepared since we saw these as supporting our core claim (as discussed). In fact we also very much appreciate your other suggestions including theoretical and empirical comparison against additional baselines, and your suggested experiment. Future iterations of this paper or follow-on work will benefit from these suggestions. Thank you once again for your engagement, your constructive feedback, and your encouragement.

---

### Official Review · Reviewer_j1se · 2024-10-30

**Soundness:** 2
**Presentation:** 1
**Contribution:** 1
**Rating:** 3
**Confidence:** 3

**Summary:**

The papar studies the goal misgeneralization issue in underspecified Markov Decision Processes, where the dynamics is controlled by a set of level parameters.  The paper compares two kinds of solutions to this problem: one is domain randomization, which tries to find a policy that maximize the reward over a distribution of levels, and the other one uses minimax expected regret, which find the policy that maximizes the reward under the worst case level parameter. The paper argues that minimax expected regret mitigates the goal misgeneralization issue, through both theoretical argument and experiments.

**Strengths:**

1. The paper considers both theory and experiments.
2. The experiments consider several variants of the baselines.

**Weaknesses:**

1. I am quite confused about the setting that the paper is actually considering: in the preliminary the paper introduces reward-free UMDP, which is reward function is not unknown to the learner or the learner does not observe reward signal at all. Then in the introduction of UED and domain randomization, there is a ground truth reward R that is optimzed by the learner - so seems like we are considering reward-based MDPs again. Both then in section 3 & 4, for example definition 3.1, it says reward-free UMDP again, and it is unclear to me that if the learner only observe R, or the learner does not observe any reward signal, or some other construction. Finally in the experiment section it seems to me that we go back to reward-based setting since the learners receive reward signal again.

2. If the goal for the theory section is to prove that MMER is better than DR in terms of distribution shift (goal misgeneralization), I am not sure if the theoretical result in the paper provides new results. Regardless of whether the paper considers reward-free or reward based UMDP, we can see that reward-free UMDP generalize / is harder than reward-based UMDP. Now if we consider reward based UMDP, and consider horizon=1, and the level parameter only controls the inital state distribution, then we can think of the control parameter is just the (distribution of) covariate in supervised learner. And the difference between optimizing for average case vs. worse case is well-known in distribution/covariate shift literature so it seems to me that the conclusions in section 4 is already implied by previous covariate shift literature, if not already existed in RL?

**Questions:**

In definition 4.1, are the states part of the trajectory?

---

> ### Author Response · Authors · 2024-11-21
>
> Thank you for your review and for your feedback and questions\! We respond to each question and point of feedback in detail as follows.
>
> **Clarification: Minimax expected regret is not the same as worst-case optimization.** In your summary you describe our approach as to “maximize reward under the worst case level parameter”. This is not accurate—we apologize for not being clear about this. There is a subtle but important distinction between minimax regret optimization and worst-case return optimization.
>
> In minimax regret, we pursue a policy that achieves the best (lowest) worst-case *regret* (maximum obtainable return minus obtained return), not the best (highest) worst-case return. This difference is important for example because a worst-case return objective will not distinguish between levels where the policy is sub-optimal (bad return and bad regret) and levels that just have very low value, for example because they are unsolvable (bad return, but good regret). This distinction is discussed in more depth by [Dennis et al. (2020, section 3\)](https://proceedings.neurips.cc/paper/2020/hash/985e9a46e10005356bbaf194249f6856-Abstract.html).
>
> Once again we apologize for not making this distinction more explicit in the paper. We have added an explicit note on line 114 in the latest revision.
>
> **Clarification: We do use reward-based MDPs.** You point out that the paper is not sufficiently clear about the use of reward-based vs. reward-free MDPs. We apologize for not making this clearer in the paper, and we appreciate you raising this issue.
>
> To clarify, our theoretical results exclusively apply to reward-based MDPs, not reward-free MDPs or any other construction. Reward-based MDPs are also the setting for all of our experiments, our RL algorithms always have access to the true reward function. Starting by introducing reward-free MDPs is merely intended as a notational convenience, since we often have a pair of environments only differing in the reward functions (but the same underlying state space, action space, and transition dynamics). Therefore it seems appropriate to talk about two reward functions and one reward-free MDP in our definitions and theorems.
>
> We have added a note to the preliminaries section (line 90\) to clarify this intention. Does this resolve your concern about this clarity issue?
>
> **On novelty:** You say that our theoretical results might be implied by previous covariate shift literature. We are committed to acknowledging relevant prior work, and we appreciate your concern. However, to the best of our knowledge, our work is novel.
>
> In this case, we are not exactly sure which literature you are referring to. Could you please point to some specific citations? Since you mention worst-case optimization, our best guess is that you may be referring to the broad literature on robust optimization, or, closer to our approach, *distributionally* robust optimization (DRO). We agree that there is a conceptual similarity between DRO and our framework, in that both involve learning on an adversarially chosen distribution. However, the DRO literature has crucial differences from our work:
>
> * **In DRO one optimizes worst-case return, which is not the same as optimizing worst-case regret** in sequential decision-making. There is work on DRO in MDPs (e.g., \[1, 2, 3\]). However, it can’t be the case that our theoretical results are implied by this prior work, because **our theoretical results only hold for worst-case regret and not for worst-case return.**
>
> * We agree that supervised learning is a special case of RL with a trivial horizon. There is much DRO work in supervised learning (e.g. surveyed in \[4\]). However, **results from supervised learning don’t generally apply in the richer setting of RL with nontrivial horizons.** Goal misgeneralization in particular is salient as a distinct generalization failure mode mainly in RL due to the central role played by reward functions as compact specifications of behavior in RL.
>
> Does this resolve your concern about the novelty of our work?
>
> References:
>
> \[1\]: [Yu and Xu, 2015](https://arxiv.org/abs/1501.07418)
> \[2\]: [Smirnova et al., 2019](https://arxiv.org/abs/1902.08708)
> \[3\]: [Derman and Mannor, 2020](https://arxiv.org/abs/2003.02894)
> \[4\]: [Kuhn et al., 2024](https://arxiv.org/abs/2411.02549)
>
> **On definition 4.1 (now 4.2):** Thanks for your question. Yes, the states were intended to be part of the state trajectory in this definition. Sorry for the unclear notation and thanks for the feedback, we have revised the definition to explicitly introduce the states as part of the trajectory. Do you think it is now sufficiently clear?
>
> **Summary:** Once again we thank you for your helpful review, which we have used to clarify and improve the paper in our latest revision. We hope that we have addressed your main concerns regarding the setting and novelty of the work. If so, we invite you to consider increasing your score.

---

> > ### Author Response · Authors · 2024-12-02
> > **Follow-up**
> >
> > Dear Reviewer j1se,
> >
> > As the end of the discussion period draws near, we wanted to quickly follow-up on your review and our response. We have attempted to address all of your stated concerns and questions about the paper:
> >
> > 1. We have clarified that we study methods of worst-case regret optimization as distinct from worst-case return optimization.
> > 2. We have clarified that we study reward-based MDPs rather than reward-free MDPs, and in our latest revision we have made this distinction explicit.
> > 3. We have outlined reasons that show that our work is distinct from prior work in distributionally robust optimization focusing on worst-case return optimization, and our results are not implied by this prior work.
> >
> > Please let us know if the above responses have addressed your concerns. Otherwise, we invite you to reconsider your rating. Thank you again for your review.

---

> > ### Comment · Reviewer_j1se · 2024-12-02
> >
> > I appreciate the reviewers for the detailed response. On the worst-case regret vs. return optimization, can we consider the following setup:
> > - let's consider cost minimization instead of reward maximization
> > - for a set of MDPs $\mathcal{M}$, we have a policy class $\Pi$ that is "realizable": for any MDP $M \in \mathcal{M}$, we have a policy $\pi^M \in \Pi$ such that $\mathbb{E}^{M,\pi^M}[c(s,a)] = 0$, where $c$ is the cost function.
> > - then can we say that in this case, the worst case regret optimization and return optimization are the same? In fact we can generatlize to any situation where the best policy from hindsight has a constant return.
> >
> > Since this is still a special case of the setup that the paper considers, we can still use the results from previous literature to prove the hardness?

---

> > > ### Author Response · Authors · 2024-12-02
> > >
> > > Thank you for your additional question.
> > >
> > > In the special case you mention, where all levels have constant maximum return, worst-case return optimization and worst-case regret optimization are equivalent.
> > >
> > > **However, this special case is overly restrictive. We consider a much broader class of problems with non-constant maximum return.** This includes any environment with $0 \< \\gamma \< 1$ and optimal trajectories with different lengths for different levels, or any environment that includes some ‘solvable’ levels as well as some ‘unsolvable’ levels. In general it is very common in reinforcement learning to consider environments that do not fall into the constant-maximum-return special case.
> > >
> > > **Worst-case return optimization is not equivalent to worst-case regret optimization outside of the constant-maximum-return special case.** Results from previous literature on worst-case return optimization that apply to that special case do not apply to the general case that we consider. For example, if we have an environment with unsolvable levels, then worst-case return optimization will prioritize unsolvable levels and the policy won’t be able to learn anything. In contrast, since these unsolvable levels have regret equal to zero, maximizing regret will not prioritize them.
> > >
> > > Therefore, we cannot use the results from previous literature in the sequential-decision making setting we consider.
> > >
> > > Please let us know if you have additional questions. Otherwise, if this addresses your remaining concern, we invite you to reconsider your rating.

---

> > > > ### Comment · Reviewer_j1se · 2024-12-03
> > > >
> > > > I think my concern still holds in two folds:
> > > > 1. Again this does not address why the negative result on UMDP without worst-case optimization is new: if supervised learning is a special case of UMDP, then the negative result in supervised learning can directly imply the hardness of UMDP without worst-case optimization. It is not clear to me why the definitions of resources and goal misgeneralization is necessary.
> > > > 2. Yes I agree that worst-case regret optimization and worst-case return optimization are different. Is there any result indicating some sense of necessity of worst-case regret optimization?

---

> > > > > ### Author Response · Authors · 2024-12-03
> > > > >
> > > > > Thanks for clarifying your remaining concern. We had misunderstood your point about the relevance of 'hardness' results from a special case. Now it's clear you're talking about the negative result in part (A) of theorem 4.1, that training with DR (maximum expected return on a fixed training distribution) performs poorly when there is only minority support on disambiguating levels in the training distribution.
> > > > >
> > > > > We agree it is plausible that there already exist similar negative results on the limitations of maximizing expected return in restricted versions of our setting. However, **this is not at all the full extent of our theoretical contribution.** To the best of our knowledge, our overall theoretical contribution is novel in the following two ways, precisely corresponding to the focus on (1) regret rather than return and (2) goal misgeneralization and resources.
> > > > >
> > > > > 1. **Regret rather than return:** Theorem 4.1 part (A) is best seen in juxtaposition with part (B) of the same theorem. Part (B) says that in the same general setting, we can achieve a strong performance guarantee from MMER (training with a worst-case regret distribution) where no such performance guarantee is possible for DR. The main takeaway of our theorem is not the DR result itself, but rather the strength of the MMER result in contrast to the DR result. To the best of our knowledge, no prior work has observed this performance guarantee of regret-based training other than in the special case that worst-case regret and worst-case return coincide (which does not imply this positive result in our general setting).
> > > > >
> > > > >     The focus on regret is necessary in the sense that, outside of this special case, theorem 4.1 (B) would not hold if we replaced worst-case regret optimization with worst-case return optimization. This is due to the differences between worst-case return and worst-case regret as we have discussed. Prior work [(Dennis et al., 2020)](https://proceedings.neurips.cc/paper/2020/hash/985e9a46e10005356bbaf194249f6856-Abstract.html) discusses the limitations of worst-case return in UMDPs in more formal detail, and their Theorem 1 shows that worst-case regret has the desirable property that in UMDPs where levels have a clear notion of 'success' and 'failure,' MMER is an optimally successful objective.
> > > > >
> > > > > 2. **Goal misgeneralization and resources:** A second aspect of our work that is novel is the setting itself, in particular our emphasis on the important problem of goal misgeneralization. Goal misgeneralization is a particular form of misgeneralization which can be framed in terms of the existence of a proxy reward function. We are the first to have connected the existence of a proxy reward function to the concept of a "disambiguating" level which naturally suggests regret-based level prioritization. In section 4.2 (was originally section 4.1) we have outlined an abstract mechanism for this "disambiguation" in terms of the existence of finite resources that place the true reward function and the proxy reward function in direct competition, leading to high regret. Work that studies misgeneralization in general (rather than goal misgeneralization in particular) cannot benefit from this approach, and in this sense our focus on goal misgeneralization is necessary.
> > > > >
> > > > > **In summary,** the focus on regret is necessary for arriving at our theorem 4.1 (B), and our focus on goal misgeneralization is necessary in that it enables our framework based on 'disambiguating levels' and our resource competition framework, which are, to the best of our knowledge, novel theoretical contributions.
> > > > >
> > > > > Once again we are grateful for your engagement during the discussion period. We hope this response addresses your remaining concern.

---

### Official Review · Reviewer_GWj5 · 2024-11-12

**Soundness:** 3
**Presentation:** 2
**Contribution:** 3
**Rating:** 6
**Confidence:** 2

**Summary:**

The paper proposes a method to mitigate goal misgeneralization in RL, which occurs when the policy generalizes capably with respect to a proxy goal but not out of distribution. The authors argue that the training signal from disambiguating levels could be amplified by regret-based prioritization. They show that approximately optimal policies on maximal-regret levels avoid the harmful effects of goal misgeneralization. Empirically, they find that current regret-based Unsupervised Environment Design (UED) methods can mitigate the effects of goal misgeneralization.

**Strengths:**

* The idea of focusing on regret-based prioritization to mitigate goal misgeneralization is novel and promising.
* A clear and well-structured theoretical analysis to support the proposed method.
* The empirical evaluation can demonstrate the effectiveness of the proposed method.

**Weaknesses:**

* How does the proposed method handle scenarios where the behavior policy's state distribution is highly sub-optimal or incomplete, potentially leading to poor coverage of desirable states?
* The structure of this paper should be more clear.
* While the authors' experiments give algorithmic insights, experiments in complex scenarios can persuade readers more easily.

**Questions:**

Please see the Weaknesses.

---

> ### Author Response · Authors · 2024-11-21
>
> Thank you for your review, for your support, and for your suggestions and questions. We are pleased that you found our approach promising as a method of mitigating goal misgeneralization, and recognised the strengths of our theoretical and empirical results. We respond in detail to each of your suggestions and questions as follows.
>
> **On sub-optimal state distributions:** We think this is a good point. We agree that current UED methods will underestimate regret in the situation where the current policy is sub-optimal according to the true reward function in disambiguating levels (e.g. where the cheese is away from the corner), such that the policy never recognises that high true return is possible (e.g. never runs into the cheese).
>
> As for how our proposed method will handle this case, we think current UED methods such as those studied in our experiments may suffer. Regret estimation is a challenge for current UED methods ([Rutherford et al., 2024](https://arxiv.org/abs/2408.15099)), and the estimators we use are limited by the performance of the current policy and are not ideal. But, this does not fundamentally limit current UED methods due to their ability to form a curriculum, and we also expect that it will be even less of a limitation for future UED methods with future advances in regret estimation. In more detail:
>
> * **As long as there are *some* levels where a UED method can recognise high regret, UED can automatically create a curriculum.** Assume there are some disambiguating levels where the current policy never achieves high return, so that regret is underestimated. Assume there are also disambiguating levels where the state distribution does allow sometimes achieving high return (e.g. levels where the cheese is a small step away from the policy’s path to the corner, such that the policy’s entropy causes it to sometimes step into the cheese). In this case, UED can automatically construct a curriculum out of progressively more ‘difficult’ disambiguating levels (e.g. by moving the cheese further and further away from the policy’s default path). Thus UED can gradually shift the policy’s state distribution to handle harder disambiguating levels.
>
> * **Our theory and empirical results suggest that advances in UED methods will lead to better mitigation of goal misgeneralization.** As the field of UED advances, we expect that better methods of estimating regret will become available. In turn, based on our theoretical results, we expect these methods will improve the ability of UED methods to mitigate goal misgeneralization.  Our experimental results already illustrate this dynamic:
>   * For example, ACCEL is a more recent UED method with a more powerful adversary than Robust PLR, and so naturally it is more capable of crafting a curriculum.
>   * In our new experiments (see [top-level comment 2/2](https://openreview.net/forum?id=po67tkP0Jx&noteId=E5lRQeVyqL) & appendix E) we also see that replacing the maximum return estimate with the ground truth optimal return (known in these environments) leads to a further increase in performance, showing that there is plenty of room for improvement from better regret estimation techniques.
>
> We appreciate your comment which has prompted us to include a brief discussion of these points in appendix E in the latest revision. Does this resolve your concern?
>
> **On clarity of the paper structure:** Thank you for the feedback. We have made some changes to the theory and results sections. We think we have substantially streamlined the theory sections in particular. Please see the [top-level comment 2/2](https://openreview.net/forum?id=po67tkP0Jx&noteId=E5lRQeVyqL) and the revised paper. Have we addressed your concern? We also welcome further suggestions.
>
> **On additional experiments:** Thank you for the suggestion. We have added new experiments in a more complex environment, please see the [top-level comment 1/2](https://openreview.net/forum?id=po67tkP0Jx&noteId=6ICGCI2do0) and appendix F in the revised paper. In summary, these new results show that our comparison between UED methods and DR continues to hold in the more complex setting. Does this address your concern?
>
> **Summary:** Thanks again for your support of the paper and for the questions and suggestions. Please consider our latest revisions and our responses to each of the suggestions and questions. We are happy to discuss any remaining doubts. If we have successfully addressed your concerns, we invite you to consider increasing your score to further support the paper.

---

> > ### Author Response · Authors · 2024-12-02
> > **Follow-up**
> >
> > Dear Reviewer GWj5,
> >
> > As the discussion period is drawing to a close, we wanted to quickly follow-up regarding your review and our response. We feel we have addressed most of your stated concerns about the paper:
> >
> > 1. Regarding sub-optimal state distributions, we have clarified the performance of UED-based methods in these cases in theory and in practice.
> > 2. Regarding the clarity of the paper, we have made revisions based on feedback from Reviewer DDwf.
> > 3. Regarding more complex environments, we have added a new appendix with additional experiments.
> >
> > Could you please confirm whether the above has addressed your concerns? If so, we invite you once more to consider strengthening your recommendation and share any follow-up questions. Thank you again for your consideration.

---

### Author Response · Authors · 2024-11-21
**Top-level comment (part 1 of 2)**

**Summary of reviews and responses:** To all our reviewers, thank you for your time and attention. In summary, you have noted several strengths of our paper:

* Reviewer DDwf noted that goal misgeneralization is an “important” and “highly relevant” problem in RL, while T3jU noted it is “interesting and important.”
* Reviewer DDwf found our framework in terms of level ambiguity novel and “very intuitive for understanding the goal misgeneralization problem.”
* You have noted that our proposed approach of applying regret-based training to tackle goal misgeneralization is “novel and promising” (GWj5) / “a novel approach” (DDwf).
* Reviewers GWj5 and j1se appreciated that we offer both theoretical and also empirical support for the potential of this approach, and DDwf praised the paper’s “strong theoretical rigor.”

However, ultimately reviewers j1se, DDwf, and T3jU reached the conclusion that the paper should be rejected. You all detailed a range of concerns with the paper including unclear presentation of the theory and experiments (DDwf), the lack of experiments in complex environments (GWj5, DDwf), and the relationship to prior work in either supervised learning (j1se) or constrained RL (T3jU), among other concerns. You also raised a number of good questions.

We thank you all for your patience while we prepared our responses and a revised version of the paper. We have taken on board several of your concerns regarding the clarity of our presentation and additional experimental evidence could better support our empirical claims, and we have submitted a revised manuscript with several new experiments and improvements in the presentation of our results.

We have responded in detail to each of your concerns and questions, or asked for further clarification, in our individual replies, detailing how we have updated the paper where relevant. In the remainder of this top-level comment, we detail the changes to the paper related to concerns shared by multiple reviewers.

---

**Summary of revision:** We are uploading a revised version of the paper with some changes.

* The three most substantial changes are the new experiments and refactored theoretical sections detailed below.
* There are a number of other clarificatory changes and minor corrections made in response to suggestions from individual reviewers, detailed in the individual responses.
* We have used our own judgment to make a small number of other local changes throughout the paper in order to clarify our work and comply with the page limit. These unnoted changes have preserved the meaning and claims from the originally submitted paper.

Select changes are highlighted in green to make them easier to spot.

We welcome follow-up questions on any of these changes.

---

**New experiments 1: More complex environment (keys and chests, appendix F):** Reviewers GWj5 and DDwf noted how the empirical contribution could be greatly strengthened by positive results in more complex environments. We conducted additional experiments in a more complex environment called ‘keys and chests.’ In short, UED methods mitigate goal misgeneralization in this environment as well. A summary of the experiment follows. For full details, please see appendix F in the revision.

* The environment is similar to the ProcGen environment ‘Heist’ adapted to demonstrate goal misgeneralization by Langosco et al. (2022) under the name ‘Keys and Chests’. The distribution shift is from levels with a small number of keys, where the policy learns to prioritize key collection, to levels with a large number of keys, where such prioritization leads the policy to largely ignore the truly rewarding chests.

* We compared DR, Robust PLR, and ACCEL (with a mutation distribution similar to the binomial mutators considered in the original experiments).We found that, while the regret estimation techniques used in current UED methods do not perform well in this environment, UED methods do successfully mitigate goal misgeneralization when using a custom environment-specific regret estimator. Under these conditions, we found that most training distributions cause DR to misgeneralize. We found that in some cases Robust PLR is able to mitigate goal misgeneralization. **In all cases tested, ACCEL was able to mitigate goal misgeneralization.**

We find these results consistent with our claim that MMER-based UED methods are a principled approach to mitigating goal misgeneralization, especially as more powerful regret estimation techniques become available. Note that developing better regret estimation methods is an ongoing research area within UED. The next set of new experiments also touches on this point.

---

> ### Author Response · Authors · 2024-11-21
> **Top-level comment (part 2 of 2)**
>
> **New experiments 2 (Regret estimation with oracle maximum returns, appendix E):** Reviewers GWj5 and DDwf asked great questions about the limitations of regret estimation. We agree about these limitations of currently available UED algorithms and regret estimators. Due to our theoretical results, we expect that improvements to regret estimation will lead future UED algorithms to be more effective at mitigating goal misgeneralization in our setting.
>
> Prompted by this discussion, we added appendix E containing new experiments on this topic. In our grid-world environments, we can use a more powerful regret estimator we call ‘oracle-latest’ that uses the known true maximum return of the level in place of an estimate of the maximum return. This gives an informative baseline that it is closer to an ideal MMER adversary (though the UED methods are still variously limited in their ability to explore the space of level distributions).
>
> We compare each UED method’s performance with oracle-latest estimation vs. max-latest estimation. The results are as hypothesized: the oracle-based methods outperform their non-oracle counterparts in all the scenarios we considered. This shows that there is ample room for performance increases derived from improvements in regret estimation. Please see our new appendix E for further discussion.
>
> ---
>
> **Refactoring the theory:** Reviewer DDwf suggested several ideas for significantly improving the presentation of our theoretical contributions. Since reviewer GWj5 expressed concerns about the structure of the paper and t3jU also had several questions about the model of competition for limited resources, we thought these changes may also interest them. The main changes are summarized as follows.
>
> * **Unified framework:** We have updated the definitions of ambiguating levels, disambiguating levels, and the proxy-distinguishing distribution shift in section 3\. The definitions now directly capture the conditions of the revised main theorem, while still accurately describing the experiments.
> * **Reordered subsections:** We have transposed the order of sections 4.1 and 4.2:
>   * Section 4.1 now contains our main result that MMER mitigates goal misgeneralization and DR may not in situations where goal misgeneralization has negative consequences (was: theorem 4.6 in section 4.2).
>   * Section 4.2 now describes our model of the competition between reward functions for limited resources, which gives a sufficient condition under which goal misgeneralization has negative consequences.
> * **Streamlined main theorem statement** (was theorem 4.6 in section 4.2, now theorem 4.1 in section 4.1):
>   * We adopted the excellent suggestion from reviewer DDwf to streamline the statement of the main theorem by expressing the assumptions directly in terms of ambiguating and disambiguating levels (section 3), rather than in terms of the existence of resources. This removes an unnecessary dependency between the two theoretical sections and better unifies the theoretical work with the experiments.
>   * We also expressed the conclusion of the theorem in terms of a defined set of approximate DR solutions and a set of approximate MMER solutions, these sets now being clearly defined in the preliminaries section, rather than being defined in-line within the statement of the theorem. This significantly shortened the length and complexity of the theorem statement.
> * **Clarified resource framework role:** In response to questions from reviewer t3jU we have clarified the framing of section 4.2 (was section 4.1) as aiming to provide an abstract condition on an environment level under which goal misgeneralization has negative consequences.
>
> ---
>
> **Summary:** We thank you again for these suggestions and questions. We think they have helped us to greatly improve the strength of the theoretical and empirical contributions. We are looking forward to follow-up discussion with all reviewers during the remainder of the discussion period.

---

### Meta-Review · Area_Chair_LFDw · 2024-12-20

**Metareview:**

The paper studied the goal misgeneralization problem, proposed an approach based on the minimax expected regret (MMER), and empirically and theoretically demonstrated that it can help mitigate goal misgeneralization.

The weaknesses of the paper are in its presentation of the theoretical results (multiple reviewers pointed that out) and its insufficient comparison to prior work (one reviewer suggested reasonable experiments to run).

**Additional Comments On Reviewer Discussion:**

The reviewers all acknowledged the authors' efforts during the rebuttal phase. However, the reviewers in general are not satisfied by the rebuttal. To summarize, one reviewer still thinks that the negative results provided in the paper are not sufficiently new (the prior known supervised learning results subsume it). Multiple reviewers raised concerns about the presentation of the theoretical results. One reviewer suggested reasonable additional experiments to compare the proposed approach and the DR baseline. The authors did not address these concerns to the level where all reviewers were convinced.

---

### Decision · Program_Chairs · 2025-01-22

Reject